# EVALUATING THE ZERO-SHOT ROBUSTNESS OF INSTRUCTION-TUNED LANGUAGE MODELS

## ABSTRACT

*Instruction fine-tuning* has recently emerged as a promising approach for improving the zero-shot capabilities of Large Language Models (LLMs) on new tasks. This technique has shown particular strength in improving the performance of modestly sized LLMs, sometimes inducing performance competitive with much larger model variants. In this paper we ask two questions: (1) How sensitive are instruction-tuned models to the particular phrasings of instructions, and, (2) How can we make them more robust to such natural language variation? To answer the former, we collect a set of 319 English instructions manually written by NLP practitioners for over 80 unique tasks included in widely used benchmarks, and we evaluate the variance and average performance of these instructions as compared to instruction phrasings observed during instruction fine-tuning. We find that using novel (unobserved) but appropriate instruction phrasings consistently degrades model performance, sometimes substantially so. Further, such natural instructions yield a wide variance in downstream performance, despite their semantic equivalence. Put another way, instruction-tuned models are not especially robust to instruction re-phrasings. We propose a simple method to mitigate this issue by introducing "soft prompt" embedding parameters and optimizing these to maximize the similarity between representations of semantically equivalent instructions. We show that this method consistently improves the robustness of instruction-tuned models.

## 1 INTRODUCTION

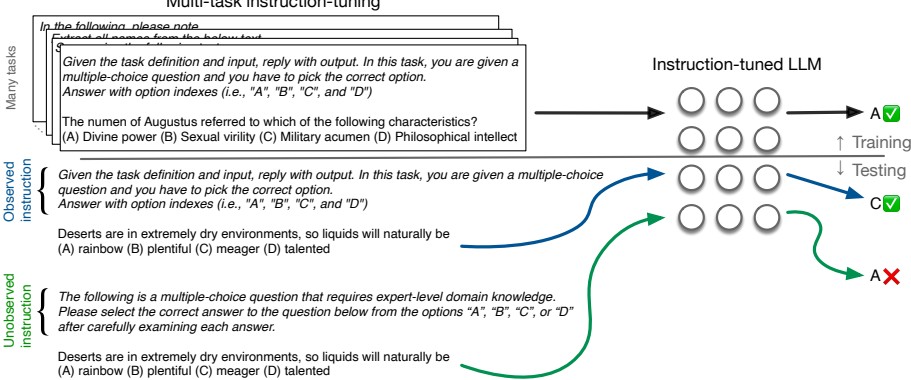

Figure 1: How well do models trained on instruction-tuning datasets generalize to novel instructions (unobserved in training)? Our analysis suggests that they do not do so very well. Shown is a case where pairing an example with an observed instruction yields the correct output, while providing a distinct but equivalent instruction produces an incorrect response. We analyze this fragility in-depth, and based on this analysis we propose and evaluate a simple method that mitigates it.

Large Language Models (LLMs) now dominate NLP, in part because they enable zero- and few-shot adaptation to new tasks via *prompting* (Brown et al., 2020; Chowdhery et al., 2022; Hoffmann et al., 2022; Zeng et al., 2022). Recent work has shown the promise of *instruction-tuning*, or fine-tuning

LLMs with natural language instructions. This improves LLM performance in zero- and few-shot settings, sometimes dramatically, especially "mid-sized" models (Chung et al., 2022; Ouyang et al., 2022). For example, the instruction-tuned Flan-T5-XL (3B parameters; Chung et al. 2022) outperforms GPT-3 (175B) on several benchmarks, despite being dramatically smaller. Similarly, Alpaca (Taori et al., 2023)—an instruction-tuned LLaMa-7B Touvron et al. (2023b)—bests GPT-3 on multiple NLP tasks. This has motivated curation of instruction-augmented task collections (Wang et al., 2022b; Wei et al., 2021), and research into improving instruction-tuning (Longpre et al., 2023; Xu et al., 2022; Sanh et al., 2021). In this work, we ask: How sensitive are instruction-tuned LMs to natural variations in instruction phrasings? This is important given that the primary motivation of instruction-tuning is to permit zero-shot adaptation to new tasks via natural language: If models are overly sensitive to particular phrasings of instructions, this may greatly limit their utility in practice.

Prior work has established that LLMs do not seem to intuitively "understand" prompts (Webson & Pavlick, 2022; Jang et al., 2023; Zhang et al., 2023a), but these efforts did not focus on instruction-tuned models and their sensitivity to plausible (appropriate) instruction phrasings. Recent, contemporaneous work (Gu et al., 2023) investigated the robustness of instruction-tuned models, and found that instruction-tuned T5 (Raffel et al., 2020) is robust to instruction perturbations in few-shot settings, but less so in zero-shot application. We contribute a more in-depth analysis of this phenomenon across a much wider set of instruction-tuned models and benchmarks. Based on this analysis, we also introduce and evaluate a method for improving the robustness of such models.

We collect a large set of task instructions manually composed by NLP researchers; these are valid instructions but distinct from those used in instruction tuning corpora. We then assess the performance of LLMs fine-tuned on different instruction corpora when provided novel (but appropriate) instructions on two large benchmarks: MMLU (Hendrycks et al., 2020) and BBL (Srivastava et al., 2022). We find that using novel instructions in zero-shot application degrades accuracy considerably (Figure 1). For example, for Flan-T5 XXL, using instructions not observed in training but which are appropriate for tasks leads to a 6.9 point drop in performance on average across benchmarks.

Our **main contributions** are summarized as follows. (1) We perform a comprehensive and in-depth analysis of the robustness of instruction-tuned LLMs across three "families" of such models (Flan-T5, Alpaca, and T0) using large benchmarks. For this we collect a large set of new task instructions composed by researchers in NLP; we release this dataset to facilitate additional work on instruction robustness. We observe substantial performance degradation when using "novel" (unseen in training) instructions. (2) We propose a simple method to improve robustness by encouraging LLMs to induce similar representations for semantically equivalent instructions; this yields promising results.

## 2 RELATED WORK

**Multitask learning and instruction-tuning**. Training a single text-to-text model capable of handling arbitrary queries has been an aspiration in NLP for at least half a decade. Prior to prompting and instructing LLMs, there were efforts to unify disparate tasks by reframing them as instances of general *question answering* (McCann et al., 2018; Khashabi et al., 2020; Keskar et al., 2019). Recent efforts have focussed on compiling and fine-tuning LLMs on corpora comprising diverse tasks with associated natural language instructions (Wei et al., 2021; Mishra et al., 2021; Sanh et al., 2021), i.e., *instruction-tuning*. For example, Wang et al. (2022b) compiles over 1600 tasks and enriches these with both instructions and negative examples. Similarly, the recently released OPT-IML Bench Iyer et al. (2022) comprises 2000 NLP tasks. The Flan 2022 task collection Longpre et al. (2023) additionally features *Chain-of-Thought* (CoT) style "reasoning" chains in instruction templates.

These meta-resources—collections of instructions, tasks, and samples—have facilitated the training of instruction-tuned model families such as Flan-T5, Flan-PaLM, and OPT-IML (Iyer et al., 2022).

**Evaluating prompting and instruction capabilities**. Instructions can be seen as a special sort of model prompting, which a few recent efforts have critically evaluated. Webson & Pavlick (2022) asked whether models meaningfully "understand" prompts, finding that they largely do not: Performance is often unaffected when irrelevant and misleading prompts are provided. In follow up work, Jang et al. (2023) evaluated performance on negated prompts and observed an "inverse-scaling" phenomenon where performance seems to degrade with model size. PROMPTBENCH (Zhu et al., 2023) provides a set of automatically generated datasets to assess the adversarial robustness of instruction-tuned LLMs. Similarly, Li et al. (2023) evaluated LLMs under adversarial instruction "attacks".

Other work has attempted to characterize when and how *in-context learning* (ICL)—i.e., including a few examples in prompts—works (Min et al., 2022; Wang et al., 2023; Dai et al., 2022; Akyürek et al., 2022; Yu et al., 2022). ICL is a form of prompting orthogonal to the present effort, as we are primarily interested in the zero-shot adaptability of instruction-tuned LLMs.

In work contemporaneous to ours, Gu et al. (2023) investigated how robust instruction-tuned models are to instruction perturbations (e.g., dropping words). This is qualitatively in line with our findings. Our work differs in important ways from this coincident research: (1) We provide a much more comprehensive analysis of robustness: Gu et al. (2023) considered *only* T5 instruction-tuned on a single dataset, whereas we evaluate three LLM families (and varying sizes of each) using five instruction tuning datasets, and we evaluate over 80 test tasks in all (Gu et al. 2023 considered only 12); from this we identify representational similarity as a source of fragility. (2) On the basis of this analysis, we propose and evaluate a new approach to *improving* the robustness of instruction-tuned models; Gu et al. (2023) offered no mechanism to improve robustness.

**Improving instruction-tuning**. Past work has also sought to improve instruction-tuning in various ways. One means to do so is to instruction tune based on human feedback (Ouyang et al., 2022; Glaese et al., 2022; Bai et al., 2022; Nakano et al., 2021; Zhang et al., 2023b). This tends to improve open-ended model responses but degrade performance on downstream tasks. Another strategy is to leverage existing resources to automatically generate instruction-tuning datasets at scale. For example, Wang et al. (2022a) use LLMs to generate instructions, inputs, and outputs and use these to improve their own instruction-following capabilities. In a similarly meta vein, Zhou and colleagues 2022 propose using LLMs to engineer prompts. Finally, Ye et al. (2022) propose "flipping" the standard task by tasking LLMs with generating *instructions*, given an input and label.

## 3 INSTRUCTION DATASETS

**Evaluation benchmarks**. We evaluate a set of instruction-tuned models on two large benchmarks—MMLU (Hendrycks et al., 2020) and BIG-BENCH (Srivastava et al., 2022)—and summarization datasets. MMLU is a multiple-choice question-answering benchmark comprising 57 tasks requiring expert knowledge. BIG-BENCH is a collaboratively built benchmark of 204 diverse tasks from various domains; we use the BIG-BENCH LITE subset, and include only QA, multi-class, binary classification tasks, and translation datasets, yielding 15 tasks in all.

**Collecting new instructions from NLP researchers**. We aim to evaluate instruction-tuned models when they are provided instructions that are appropriate for a given task but superficially different from instructions observed in training. To this end we enlisted NLP researchers (graduate students) to compose novel instructions for the tasks considered; these particular instruction phrasings were therefore *unobserved* during instruction fine-tuning.

Briefly, we recruited 36 NLP graduate students working in NLP. All had at least some experience with instruction-tuned models and the downstream tasks included in the evaluation benchmarks. For all tasks in BBL and MMLU, 12 graduate students wrote one instruction they would use for zero-shot inference with an instruction-tuned model. We provide additional details on this instruction collection process in Appendix J. We will release all 319 instructions collected for reproducibility and to facilitate further research on instruction-tuned model robustness.

## 4 EVALUATING THE ROBUSTNESS OF INSTRUCTION-TUNED LLMs

### 4.1 MODELS AND DATA

We conduct experiments with publicly accessible model variants trained over three instruction collections (which provide *observed* task instructions): P3 (Sanh et al., 2021), Flan-2022 (Chung et al., 2022), and Alpaca (Taori et al., 2023). We do not include Llama2-Chat (Touvron et al., 2023b) because its instruction-tuning data is not available. But we do offer a comparison between LLama2-Alpaca and LLama-Alpaca in Appendix D (which suggests that better pre-training does not markedly improve robustness). We manually identified all instructions that correspond to the following task types: (a) Multiple-choice question answering (QA); (b) Binary classification (BC); (c) Multi-class classification (MC); (d) Summarization and; (e) Translation.

| | |
|---|---|
| QA | In this task, you are given a multiple-choice question and you have to pick the correct option. Answer with option indexes (i.e., "A", "B", "C", and "D"). Q: {question} A. {choiceA} B. {choiceB} C. {choiceC} D. {choiceD} |
| MC | Pick one category for the following text. The options are - {options} {text} |
| BC | {paragraph} Choose your answer: According to the above paragraph, the question "{question}" is "{response}"? |

Table 1: Examples of observed instructions we collected for three general types of tasks.

| OBSERVED INSTRUCTIONS | | | | | | UNOBSERVED INSTRUCTIONS | | |
|---|---|---|---|---|---|---|---|---|
| *Instruction Type* | QA | MC | BC | Sum | Trans | Number of tasks | 1 | 15 |
| Flan | 50 | 35 | 18 | 37 | 14 | Instructions per task | 20 | 10 |
| Alpaca | 20 | 20 | 11 | 20 | 10 | Total instructions | 20 | 140 |
| P3 | 13 | 8 | 7 | 24 | - | | | |

Table 2: Counts of instruction phrasings (unobserved and observed) we use for evaluations.

To evaluate model robustness to instruction phrasings, we use two large benchmarks: MMLU and BIG-BENCH LITE (BBL) along with the acquired set of novel instructions described above.

We include all 57 tasks from MMLU and 14 of 24 tasks from BBL.[1] We use the same instructions for all tasks in the same category, taken from the instruction tuning datasets associated with each model. These instructions are general, e.g., in the case of classification they request that the model consider an example with respect to categorization criteria and label space provided by the instance, and select an appropriate category (examples in Table 1). One can "mix-and-match" such instructions so long as they are appropriate for the task type.

## 4.2 MAIN RESULTS

We present the main aggregated analysis results in Figure 2 and Table 3. The take-away here is that using instructions unobserved in training—but manually composed for the task at hand and so semantically appropriate—leads to considerable degradation in performance: On average, unobserved instructions reduce accuracy by over five points across models considered. Table 3 reports results disaggregated by task type; we observe that classification tasks are most harmed by use of novel instructions. We provide additional, more granular (dataset-level) results in Appendix K.

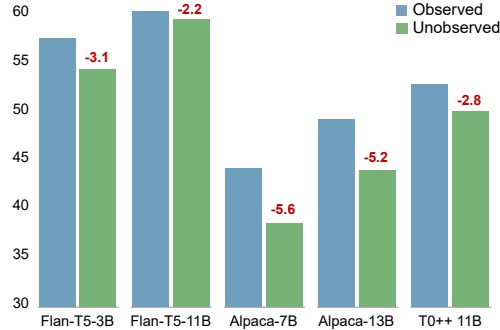

(a) Average zero-shot performance over all tasks when using observed and unobserved instructions.

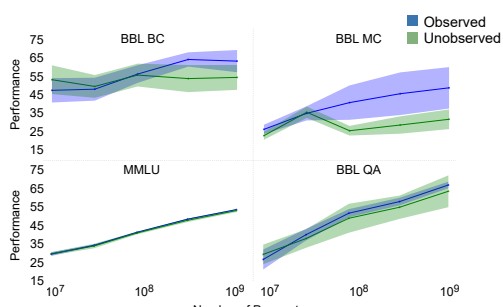

(b) Performances of Flan-T5 using observed and unobserved instructions as a function of model size.

Figure 2: Using novel but valid instructions at test time (phrasings unobserved in training) consistently degrades the performance of instruction-tuned LLMs (a). Scale does not necessarily fix this (b).

We present results for NLG tasks (translation and summarization) in Table 14. For these we use ROUGE (Lin, 2004), BLEU (Papineni et al., 2002), and BERTScore (Zhang et al., 2019) as automatic

---

[1]We exclude 4 BBL exact-matching tasks, and 5 including tokens not recognized by T5 (e.g., emojis).

| Model | MMLU | | BBL-QA | | BBL-BC | | BBL-MC | | **Overall** | |
|---|---|---|---|---|---|---|---|---|---|---|
| | Avg. | Std. | Avg. | Std. | Avg. | Std. | Avg. | Std. | Avg. | Std. |
| Flan-T5-3B | | | | | | | | | | |
| Observed | **48.1** | (±0.3) | **59.0** | (±2.1) | **66.5** | (±3.8) | **55.6** | (±0.7) | **57.3** | (±1.7) |
| Unobserved | 47.5 | (±0.9) | 56.0 | (±7.3) | 61.1 | (±6.9) | 52.1 | (±5.4) | 54.2 | (±5.1) |
| **Performance △** | ↓ 0.6 | | ↓ 3.0 | | ↓ 5.5 | | ↓ 3.5 | | ↓ 3.1[†] | |
| Alpaca-7B | | | | | | | | | | |
| Observed | **41.9** | (±0.6) | **48.6** | (±2.8) | **53.8** | (±3.4) | **32.1** | (±2.2) | **44.1** | (±2.3) |
| Unobserved | 39.7 | (±2.2) | 45.3 | (±6.5) | 52.4 | (±6.5) | 16.4 | (±3.5) | 38.5 | (±4.7) |
| **Performance △** | ↓ 2.2 | | ↓ 3.3 | | ↓ 1.4 | | ↓ 15.7 | | ↓ 5.6[†] | |
| T0++ 11B | | | | | | | | | | |
| Observed | 48.3 | (±0.9) | 54.1 | (±4.1) | **66.1** | (±2.1) | **42.0** | (±2.1) | **52.6** | (±2.3) |
| Unobserved | **48.5** | (±0.9) | **54.7** | (±3.7) | 54.7 | (±4.3) | 41.4 | (±2.4) | 49.8 | (±2.8) |
| **Performance △** | ↑ 0.2 | | ↑ 0.7 | | ↓ 11.4 | | ↓ 0.6 | | ↓ 2.8 | |
| Flan-T5-11B | | | | | | | | | | |
| Observed | **53.2** | (±0.2) | **67.9** | (±1.8) | **65.6** | (±6.0) | **58.7** | (±0.5) | **61.4** | (±2.1) |
| Unobserved | 52.7 | (±0.8) | 64.6 | (±8.5) | 63.6 | (±6.1) | 55.9 | (±5.5) | 59.2 | (±5.2) |
| **Performance △** | ↓ 0.5 | | ↓ 3.4 | | ↓ 2.0 | | ↓ 2.8 | | ↓ 2.2[†] | |
| Alpaca-13B | | | | | | | | | | |
| Observed | **47.8** | (±0.5) | **53.9** | (±2.2) | **57.9** | (±4.8) | **36.7** | (±1.8) | **49.1** | (±2.3) |
| Unobserved | 47.0 | (±0.8) | 51.7 | (±5.7) | 54.1 | (±5.6) | 22.7 | (±7.5) | 43.9 | (±14.0) |
| **Performance △** | ↓ 0.9 | | ↓ 2.2 | | ↓ 3.8 | | ↓ 14.0 | | ↓ 5.2[†] | |

Table 3: Results using observed and unobserved instructions across benchmark tasks (grouped by type). Performance degrades—sometimes by 10+ points—when one uses (Unobserved) instructions, suggesting that instruction-tuned models are not particularly robust. BC, MC, and QA stand for binary classification, multi-class classification, and question answering, respectively. A [†] denotes a statistically significant difference ($p < 0.05$) over datasets for a given model under a paired $t$-test.

metrics. To ensure a fair comparison, we evaluate only examples comprising <512 tokens (including instructions). We use $61.5\%$ test set data for XSUM (Narayan et al., 2018), $17.6\%$ for CNN/DM (Nallapati et al., 2016), and $4.0\%$ for MEDIASUM (Zhu et al., 2021).

| Model | **Held-in Datasets** | | | | **Held-out Datasets** | | | |
|---|---|---|---|---|---|---|---|---|
| | CNN/DM $R_2$ | | XSUM $R_2$ | | MEDIASUM $R_2$ | | WMT19 FR-DE $B_2$ | |
| | Avg. | Std. | Avg. | Std. | Avg. | Std. | Avg. | Std. |
| Flan-T5-3B | | | | | | | | |
| Observed | **41.5** | (±0.1) | **36.1** | (±1.0) | **23.4** | (±6.2) | **19.2** | (±1.8) |
| Unobserved | 32.3 | (±2.4) | 35.7 | (±1.1) | 18.0 | (±1.4) | 18.4 | (±3.0) |
| **Performance △** | ↓ 9.1 | | ↓ 0.4 | | ↓ 5.4 | | ↓ 0.7 | |
| Alpaca-7B | | | | | | | | |
| Observed | **19.2** | (±3.4) | **12.4** | (±1.3) | **9.9** | (±1.6) | **16.5** | (±6.8) |
| Unobserved | 16.8 | (±0.9) | 8.3 | (±0.4) | 9.1 | (±0.7) | 16.2 | (±6.9) |
| **Performance △** | ↓ 2.4 | | ↓ 4.1 | | ↓ 0.7 | | ↓ 0.3 | |

Table 4: Results using observed and unobserved instructions on NLG tasks. We measure the performance gap on both held-in datasets and held-out datasets for instruction-tuning. Performance is again worse when unobserved instructions are used. $R_2$ refers to ROUGE-2 F1 and $B_2$ refers BLEU with $n$-gram. These are representative results; detailed results with more metrics are in Appendix B

## 4.3 DOES INSTRUCTION ROBUSTNESS EMERGE AS A FUNCTION OF SCALE?

We repeated all experiments from Table 3 with Flan-T5 model sizes ranging from small (80M parameters) to XXL (11B). Figure 2b shows that the disparity between results achieved with observed versus unobserved instructions **does not** decrease with model scale, at least up to this point. That said, massive models (175B+) may offer improved robustness. However, we reiterate that much of

the excitement about instruction-tuning that this technique appears to allow much smaller models to achieve results competitive with massive alternatives. We report more details in Appendix H

## 4.4 ROBUSTNESS UNDER IN-CONTEXT LEARNING (ICL)

Recent work (Gu et al., 2023) showed that LLMs are less sensitive to prompt variation when provided "few-shot" examples in context. We are focused on zero-shot, instruction-following capabilities; but for completeness we re-ran all experiments in a few-shot setting. We followed the sub-sampling strategy from 4.2, and we evaluate the same examples in the few-shot settings we did for the zero shot case. Given space constraints, we report the full results in Appendix C, but briefly summarize our findings here briefly. (1) On average, ICL examples improve both **performance** (avg. performance) and **robustness** (i.e., performance with novel compared to observed instructions) slightly for most models and tasks. (2) For each task, the *variance* of **performance** and **robustness** significantly increases when more shots are provided, suggesting a sensitivity to the specific ICL examples selected. In particular, model **robustness** varies significantly with choice of ICL examples.

In sum, ICL does mitigate the robustness issue of medium-sized LLMs we have highlighted, but this is **unstable** and depends on the particular selection of examples.

## 4.5 A CLOSER LOOK AT INSTRUCTION ROBUSTNESS

Above we used general instructions to induce LLMs to perform tasks (Table 1). Here we delve further into the performance degradation observed when using novel instructions. We report a curious result: *Incorrect but observed instructions outperform appropriate but unobserved instructions* (Figure 3).

We come to this observation by evaluating the performance of Flan-T5-XXL (11B) using six instruction types over seven datasets from BIG-BENCH. This includes (variants of) two instructions *observed* in training: **Closest** is the instruction from the most similar task in the instruction-tuning set; **Incorrect** is an observed instruction for a *completely different* and inappropriate task (but which has the same desired output format)—intuitively, these should not yield the desired behavior; **Negated** is the same as **closest**, but we negate the instruction to indicate that it should *not* perform the task. For *unobserved* instructions, we consider: **Task designer**, the instruction (task prefix) provided by the author of the task in BIG-BENCH, and; **Newly collected**, novel and appropriate instructions we collected as described above. For reference, we also consider **Nonsensical**, which is a random "instruction" completely irrelevant to any task.

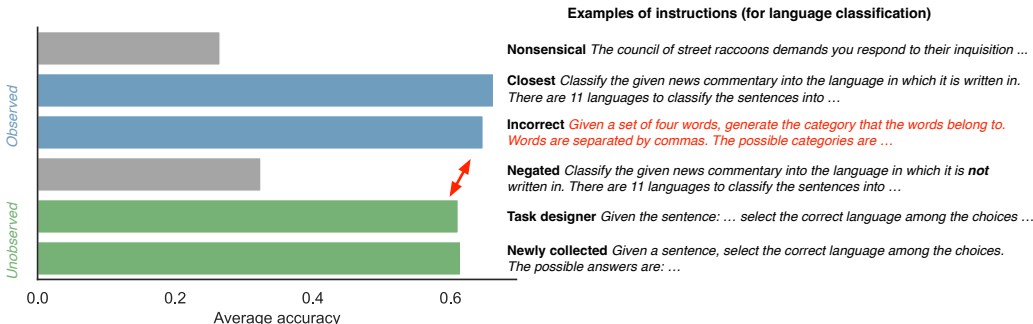

Figure 3: *Incorrect* **but observed instructions perform better on average than** *correct* **but unobserved instructions**. We report averages over benchmarks, but show example instructions on the right for a specific, illustrative task. We provide all instructions in the Appendix.

Figure 3 reports average results for these variants. Consistent with our findings, using instructions unobserved in training degrades performance. Strikingly, here we also find that using *inappropriate but observed* instructions outperforms using *appropriate but unobserved* instructions. This indicates that instruction-tuned models—at least the modestly sized ones we have evaluated here—may overly rely on having observed instructions in training, and not generalize to new instructions and phrasings as we might hope. We provide a full set of disaggregated results in Appendix B.2.

### 4.6 ROBUSTNESS AND "SEMANTIC" DISTANCE

One observation in 4.2 is that performance on MMLU is less affected by using unobserved instructions. MMLU is a benchmark with 57 QA tasks covering different domains; these tasks all share a similar form of input-output (question, four choices → answer). During instruction collection, we treated all tasks in MMLU as a general QA task and asked NLP researchers to write general QA instructions. We therefore hypothesize that these instructions are comparatively similar to observed instructions, and this in turn explains the relative robustness in this case.

Figure 4 and Table 5 support this hypothesis. For each instance (instruction plus example), we extract the representation at the penultimate layer for the first decoded token. We use tSNE (Van der Maaten & Hinton, 2008) to visualize these representations of observed and unobserved instructions over instances in MMLU and BBL. Figure 4 shows that for MMLU, the unobserved instructions we collected are quite similar to the observed, while there is a greater separation between unobserved and observed instructions in BBL. Table 5 further quantifies this phenomenon: We report the average $\ell 2$ distance between representations of unobserved instructions and those of their nearest observed counterparts. MMLU unobserved instructions are, on average, closer to the nearest observed instruction; this correlates with the smaller observed performance drop.

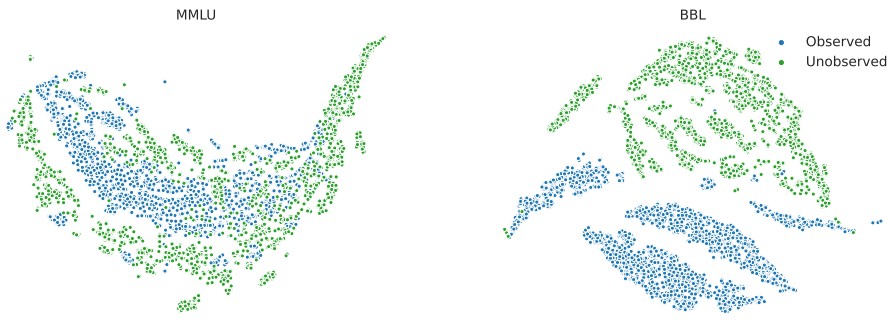

Figure 4: tSNE plots of representations for the first decoded tokens of 300 randomly sampled examples from MMLU and BBL with Flan-T5 (XXL). Embeddings of observed and unobserved instructions for MMLU are similar, while for BBL they are quite different. This result holds across **most but not all** models considered: See Appendix I for visualizations over all models.

We plot mean performance degradation (as %) as a function of average similarity between the similarity of the first decoded tokens (following *unobserved* instructions) and the same for the *most similar observed* instruction. The negative slope implies the intuitive relationship: Instructions that are dissimilar (in terms of model representations) correlate with poorer performance. However, the relationship is relatively weak, yielding a slope estimate of -0.2 ($p =$0.08).

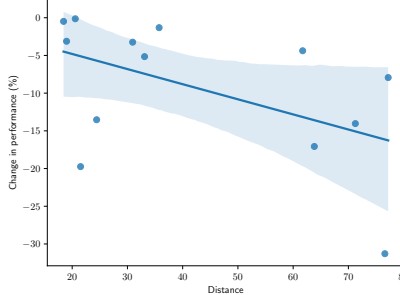

| Dataset | Avg. $\Delta \ell 2$ | Avg. $\Delta$ Acc. |
|---------|---------|---------|
| MMLU | **19.8** | **-0.5** |
| BBL-QA | 37.9 | -3.4 |
| BBL-BC | 25.3 | -2.0 |
| BBL-MC | 26.1 | -2.8 |

Figure 5: Average degradations in performance versus "semantic" distance while using unobserved instructions. In general, larger semantic distances correspond to increased degradation.

Table 5: Average degradations in performance for four categories (Flan-T5-XXL). MMLU has minimal average distance, indicating a smaller distribution shift; we hypothesize that this is what leads to the relatively small degradation in performance.

## 5 ALIGNING REPRESENTATIONS OF EQUIVALENT INSTRUCTIONS

Based on the preceding analysis, we hypothesize that that *to be robust instruction-following LLMs should induce close internal representations for instructions corresponding to similar tasks*. Here we propose a simple approach to explicitly realize this aim: We introduce a term in the objective that encourages the model to yield similar predictions (and, correspondingly, similar representations) for the same input when provided distinct but semantically equivalent instructions.

Specifically, we add soft embedding parameters with dimensions $\mathbb{R}^{d \times n}$; this is equivalent to adding $n$ novel tokens (with embedding dimension $d$) as prefixes to inputs (preceding instructions). The intuition is to push representations for semantically equivalent tasks close together: To this end, fine-tune the soft embeddings (Li & Liang, 2021) under an additional loss term: The KL-divergence $\mathcal{L}_{\mathrm{KL}}$ of the output probabilities between a reference instruction for a given task and a paraphrased (effectively equivalent) version of the same. We combine this with the standard cross-entropy loss and fit *only* the introduced soft prompt parameters under this objective (Figure 6). Here $\lambda$ is a loss-weighting hyper-parameter, $\hat{y}_i^{(j)}$ and $\hat{y}_r^{(j)}$ are the distributions over the vocabulary $\mathcal{V}$ induced by the model with paraphrased instruction $i$ and the reference instruction $r$ at token position $j$.[2]

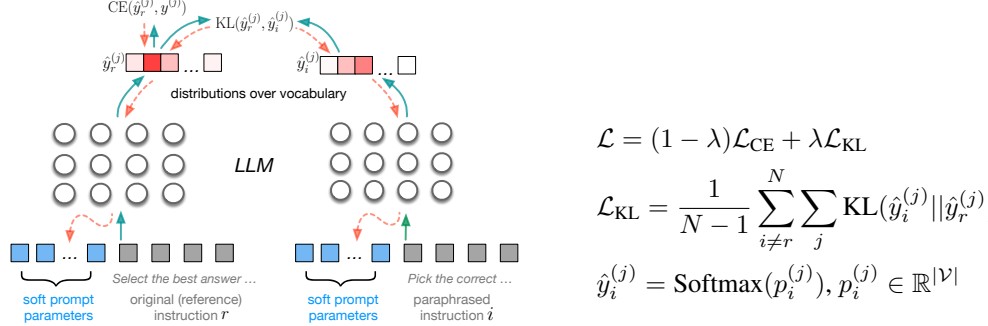

Figure 6: Schematic depiction of the proposed instruction alignment method (left) and associated loss terms (right). Dotted (red) lines indicate backpropagation; we update only the soft prompt parameters, which we show yields performance superior to fine-tuning all model parameters.

Optimizing for the above objective requires similar instructions $i$ for each task in the training data; we generate these automatically as follows. For the instruction-tuning dataset, we sample a small amount of training data to use for representation alignment. We paraphrase these reference instructions using GPT-4. For the Alpaca collection, we randomly sampled 1k tasks, paraphrased them with three prompts, and collected the top three candidates under temperature 0.5. For the Flan collection, we randomly sampled 1k instances from the mixture with 3 prompts with greedy decoding (see B.3).

For fine-tuning, we create instances for each example by pairing them with every distinct instruction available for the corresponding task. We then form batches by including one instance featuring the original instruction, and the rest comprising paraphrased instructions.

## 6 RESULTS

We implement the proposed method to push together the representations of instructions for two models: Flan-T5-XL (3B) and Alpaca (7B). After training with the added prefix, we reproduce the main results from Section 4.2. Then, following the approach in Section 4.6, we take the average distance between observed and unobserved instructions before and after. Table 6 shows that this brings observed and unobserved instruction representations closer together. The similarity is most increased in the case of the biggest accuracy gain, supporting the hypothesized mechanism of improvement. We compare the canonical versions of these models trained in the usual way (the same evaluated in Table 3) to variants fine-tuned with the proposed approach. We ablate components of our method to tease out the contributions of the extra parameters and closer representation. Specifically, we

---

[2]We pad instances such that the lengths in a given batch are effectively equal; the sum is therefore from 1 to the length associated with the current batch, we omit this for simplicity.

| Dataset | Closest Distance Before | Closest Distance After | $\Delta$ Acc. Improvement (%) |
|---------|------------------------|------------------------|-------------------------------|
| MMLU    | 22.2                   | 21.3                   | + 0.3%                        |
| BBL QA  | 22.4                   | 23.0                   | + 0.4%                        |
| BBL BC  | 30.1                   | **27.9**               | **+ 4.2%**                    |
| BBL MC  | 26.0                   | 24.6                   | + 0.3%                        |

Table 6: Average distances before and after soft prompt alignment with Flan-T5-XL.

consider variants where we: Fine-tune all model parameters on the additional, automatically generated instruction paraphrases (FT); impose the new KL loss term (again fine-tuning all model parameters; FT+KL); introduce the additional soft prompt parameters and fine-tune on the paraphrase instances, but without KL (PT); and then the full proposed strategy, which introduces the soft prompt parameters and optimizes them for the loss augmented with the KL term (**PT+KL**).

| | MMLU | | | BBL | | |
|---|---|---|---|---|---|---|
| **Model** | OBS. | UNOBS. | Avg. | OBS. | UNOBS. | Avg. |
| FLAN-T5-3B | 48.1 | 47.5 | 47.8 | **56.1** | 51.9 | 54.0 |
| FT | 39.4 (**-8.7**) | 40.1 (**-7.4**) | 39.8 (**-8.0**) | 48.2 (**-7.9**) | 42.3 (**-9.2**) | 45.3 (**-8.7**) |
| FT+KL | 41.8 (**-6.3**) | 43.6 (**-3.9**) | 45.9 (**-1.9**) | 47.7 (**-8.4**) | 43.1 (**-8.8**) | 45.4 (**-8.6**) |
| PT | 48.1 (**+0.0**) | 47.6 (**+0.1**) | 47.9 (**+0.1**) | 55.9 (**-0.2**) | 52.1 (**+0.2**) | 54.0 (**+0.0**) |
| **PT+KL**[†] | **48.1** (**+0.1**) | **47.9** (**+0.4**) | **48.0** (**+0.2**) | 55.9 (**-0.2**) | **53.7** (**+1.8**) | **54.8** (**+0.8**) |
| ALPACA-7B | 41.9 | 39.7 | 40.8 | 47.6 | 42.9 | 45.3 |
| FT | 40.3 (**-1.6**) | 39.1 (**-0.6**) | 39.7 (**-1.1**) | 44.4 (**-3.2**) | 42.1 (**-0.8**) | 43.4 (**-2.0**) |
| FT+KL | 39.7 (**-2.2**) | 40.2 (**+0.5**) | 40.0 (**-0.8**) | 45.6 (**-2.0**) | 42.8 (**-0.1**) | 44.2 (**-1.1**) |
| PT | 42.1 (**+0.2**) | 40.0 (**+0.3**) | 41.1 (**+0.3**) | 47.5 (**-0.1**) | 43.0 (**+0.1**) | 45.3 (**+0.0**) |
| **PT+KL**[†] | **42.4** (**+0.5**) | **41.8** (**+2.1**) | **42.1** (**+1.3**) | **47.9** (**+0.3**) | **46.6** (**+3.7**) | **47.3** (**+2.0**) |

Table 7: Results and ablations of the proposed soft prompt alignment method. Ablated versions use the augmented set with paraphrased instructions. FT refers to fine-tuning (with teacher-forcing) on this additional data; PT denotes prefix tuning (i.e., introducing soft prompt parameters); KL refers to the alignment objective that we proposed above. Using all components together yields the best performance, especially on unobserved instructions; [†] this improvement is statistically significant ($p < 0.05$) for both models under separate paired $t$-tests. We report variances in Appendix B.

We report results in Table 7. Two observations: (1) The proposed soft prompt alignment strategy (**PT+KL**) yields consistent improvements across the tasks and models considered and especially improves performance on unobserved instructions, as anticipated. (2) Simply fine-tuning with extra parameters or data (e.g., the newly collected paraphrases) on the same training tasks does not improve the overall performance and robustness.

# 7 CONCLUSIONS

Instruction-tuned LLMs have emerged as a promising means of achieving zero-shot performance with smaller models that are competitive to that observed using much larger LLMs. We characterized the *robustness* of such models with respect to instruction rephrasings. We collected manually composed instructions from 36 graduate students in NLP across 75 tasks, and we evaluated different families of instruction-tuned LLMs (Flan, Alpaca, and T0) when provided observed and unobserved instructions (seen in training and not, respectively). We found that using the latter consistently degrades model performance, indicating that models are unduly sensitive to instruction phrasings.

Our analysis suggested that an issue may be the internal representations of instructions: Equivalent instructions should be near each other in latent space. On this hypothesis, we introduced a simple mechanism intended to improve the robustness of instruction-tuned LLMs. This approach entails introducing an additional loss term that penalizes the model for inducing dissimilar distributions when using (a) paraphrased instructions as opposed to (b) a reference instruction for the same task. We found that training under this objective consistently improves results and mitigates the degradation observed when previously unobserved instructions are used.

## 8 ETHICS STATEMENT

This work does not have an explicit ethical dimension, but we acknowledge that all LLMs are likely to encode problematic biases; it is currently unclear how instruction-tuning might interact with these, and we think this is a topic worth pursuing in future work.

## 9 REPRODUCIBILITY STATEMENT

All the results reported in the paper are reproducible. We submit the code and include all the implementation details in Appendix B. We present the disaggregated results in Appendix K. All the instructions used in the paper are formatted and listed in the supplementary materials.

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

## A  LIMITATIONS

This work has important limitations: For example, we only evaluated "mid-sized" models (<20B parameters), it is unclear if our findings would generalize to much larger instruction-tuned models. (However, we note that instruction tuning has been most promising for smaller models.) We also considered only a somewhat limited set of task types: QA, multi-class and binary classification, and then summarization and translation for generation tasks.

## B  EXPERIMENTS

To ensure reproducibility, we provide all details regarding our evaluation of the robustness of instruction-tuned LLMs.

### B.1  MAIN RESULTS

LLMs sometimes generate outputs that are *correct* but different from a (natural language) target. Therefore, we predict answers according to "multiple-choice" grading suggested by BIG-BENCH, by which we take the logits score and argmax over all the possible choices to obtain the prediction. In most cases, this approach yields the same accuracy as using exact match for evaluation. Here are the configurations for all the models we evaluated.

| Models | Node Type | Precision | Batch Size | Hours | $CO_2$ emission (KG) |
|---|---|---|---|---|---|
| **Inference** | | | | | |
| Flan-T5-Small | V100-SXM2-32G | FP16 | 128 | 64 | 4.0 |
| Flan-T5-Base | V100-SXM2-32G | FP16 | 128 | 128 | 8.1 |
| Flan-T5-Large | V100-SXM2-32G | FP16 | 32 | 256 | 16.2 |
| Flan-T5-XL | V100-SXM2-32G | FP16 | 32 | 512 | 32.3 |
| Flan-T5-XXL | RTX-A6000-46G | BF16 | 8 | 600 | 37.8 |
| T0++ | RTX-A6000-46G | BF16 | 2 | 128 | 8.1 |
| Alpaca-7B | A100-SXM4-80G | BF16 | 16 | 160 | 13.4 |
| Alpaca-13B | A100-SXM4-80G | BF16 | 8 | 192 | 16.1 |
| **Training** | | | | | |
| Flan-T5-XL | A100-SXM4-80G | BF16 | 256 | 256 | 21.5 |
| Alpaca-7B | A100-SXM4-80G | BF16 | 128 | 80 | 6.7 |
| **Estimated Total $CO_2$ Emission (KG)** | | | | | 164.2 |

Table 8: The configurations for evaluating different instruction-tuned LMs. The $CO_2$ emission is estimated by Lacoste et al. (2019). Total estimated emissions are about equivalent to a 679 KM drive by an average ICE car.

### B.2  "CLOSER LOOK" EXPERIMENT RESULTS

We provide detailed results that correspond to the aggregated values we reported in Figure 3.

Table 9 shows that among the 7 benchmark datasets that we experiment with, the "Incorrect" instructions outperform the "Collected" instructions. And then for the other 2 datasets the performance of "Incorrect" instructions is comparable ($-1\%$) to the "collected" ones. This indicates that the instruction-following ability of the LLMs is not robust to the fine-grained semantics within the instructions, as it only cares about the high-level requirement of the prompt. (classification, QA, etc.)

#### B.2.1  MOTIVATION AND INSTRUCTION COLLECTION

Here, we report the detailed procedure and motivation for collecting the instruction templates for the six settings we experiment with.

| Dataset | Observed | | Unobserved | | Control | |
|---|---|---|---|---|---|---|
| | Closest | Incorrect | Collected | Task Designer | Negated | Nonsensical |
| Intent Recognition | 93.6 ±0.3 | 93.1 ±1.0 | 94.1 ±0.6 | **94.7** | 28.0 ±6.5 | 40.7 ±7.2 |
| Empirical Judgments | 39.2 ±0.8 | **41.6** ±6.3 | 37.6 ±1.7 | 37.4 | 28.1 ±3.5 | 30.9 ±2.5 |
| Conceptual Combinations | 78.0 ±1.6 | **78.9** ±0.5 | 75.3 ±3.3 | 58.3 | 11.2 ±2.1 | 63.7 ±2.6 |
| Language Identification | **38.9** ±0.3 | 29.3 ±5.0 | 28.8 ±5.8 | 27.6 | 36.9 ±0.5 | 12.4 ±0.5 |
| Logical Sequence | **56.9** ±6.6 | 49.4 ±6.1 | 52.8 ±5.3 | 53.8 | 11.8 ±6.9 | 34.4 ±5.9 |
| Crash Blossom | 53.6 ±2.8 | 50.0 ±5.3 | 50.5 ±2.2 | **63.2** | 28.6 ±6.2 | 43.7 ±1.4 |
| Epistemic Reasoning | 62.8 ±2.9 | 59.3 ±3.4 | 58.1 ±1.7 | 60.2 | **65.5** ±4.6 | 49.5 ±1.3 |
| Overall | **60.5** ±2.1 | 57.4 ±2.2 | 56.8 ±1.8 | 56.4 | 30.0 ±2.2 | 39.3 ±2.3 |

Table 9: The detailed results of "A Closer Look" experiment. Using "Incorrect" but observed instructions in most cases outperforms correct but unobserved instructions ("Closest" and "Collected", respectively).

**Closest Instructions**    In the main results reported in 4.2, we explicitly use the observed instruction templates that are general enough to be used for **all** the task belongs to the same categories. We show that even with such generality, the observed instructions template still outperforms the unobserved ones by a large margin. Next we evaluate whether the observed instructions for similar trained tasks perform better than the general ones. For each task, we collected the closest instructions by **manually** going through the instruction tuning set, and selecting suitable instructions from the train tasks that are similar to the evaluation tasks, e.g., instructions from WMT16 (trained) for WMT19 (evaluate).

**Incorrect Instructions**    As an extreme opposite setting to **Closest Instruction**, we investigate how the models perform when the given instructions are from completely different tasks, but have a similar input-output format (e.g., language classification and movie genre classification). This setting **should not** work if the model "understands" instruction semantics. For each task, we collect the closest instructions with the following procedure. We again **manually** go through the instruction tuning set and filter a set of tasks that belongs to the **same category** (QA, summarization, Multi-class classification, and so on). Among tasks in the same category, we identify instrctions from the most different task.

**Negated Instructions**    Following the design of Jang et al. (2023), we use negated instructions as an additional probe to assess whether the model understands instructions or mostly recalls back to the training data. For each task, we collect negated instructions by:

(1) Taking all of the **Closest Instructions** for this task.

(2) Insert a negation word (e.g., not) in the appropriate place of the instruction.

**Task Designer Instruction**    For the benchmark BBL that we evaluate, for each task, it provides a prompt template written by the task designer. This serves as a gold unobserved instruction that is guaranteed valid for this task.

**Newly Collected Instruction**    This is the same as the "Unobserved Instruction" that we used throughout the paper. Following the exact same procedure in Section 4.2, we collect 10 unobserved instructions for each task we evaluate in this section.

**Nonsensical Instruction**    This setting is used as a baseline. We perform inference of the task with an instruction prepended to the input, that is irrelevant and doesn't provide any extra information. (e.g., *"The council of street squirrel demands you response: Question: ..."*)

### B.3    DETAILS ON ALIGNING REPRESENTATIONS OF EQUIVALENT INSTRUCTIONS

We conduct all training and ablation studies on 8 A100s with 80GB memory. We kept the KL-Loss weight fixed at $0.8$. We trained both Flan-T5-XL and Alpaca-7B with a batch size of 4 and a batch gradient accumulation of 4. We set the weight decay to $1e-5$ and the learning rate to $5e-4$ for all experiments. We use a prefix length of 10. We explore the choice of the prefix length and found no significant correlation with the performance and "semantic" distance between instructions.

| | MMLU | | | BBL | | |
|---|---|---|---|---|---|---|
| **Model** | OBS. | UNOBS. | Avg. | OBS. | UNOBS. | Avg. |
| FLAN-T5-3B | 48.1 ($\pm$0.3) | 47.5 ($\pm$0.9) | 47.8 ($\pm$0.6) | **56.1** ($\pm$5.8) | 51.9 ($\pm$9.6) | 54.0 ($\pm$7.7) |
| FT | 39.4 ($\pm$0.5) | 40.1 ($\pm$0.4) | 39.8 ($\pm$0.4) | 48.2 ($\pm$5.6) | 42.3 ($\pm$8.8) | 45.3 ($\pm$7.2) |
| FT+KL | 41.8 ($\pm$0.3) | 43.6 ($\pm$0.2) | 45.9 ($\pm$0.3) | 47.7 ($\pm$4.2) | 43.1 ($\pm$7.3) | 45.4 ($\pm$5.8) |
| PT | 48.1 ($\pm$0.3) | 47.6 ($\pm$0.8) | 47.9 ($\pm$0.6) | 55.9 ($\pm$6.0) | 52.1 ($\pm$9.9) | 54.0 ($\pm$8.0) |
| **PT+KL** | **48.1** ($\pm$0.2) | **47.9** ($\pm$0.4) | **48.0** ($\pm$0.3) | 55.9 ($\pm$4.4) | **53.7** ($\pm$5.9) | **54.8** ($\pm$5.1) |
| ALPACA-7B | 41.9 ($\pm$0.6) | 39.7 ($\pm$2.2) | 40.8 ($\pm$1.4) | 47.6 ($\pm$2.8) | 42.9 ($\pm$5.5) | 45.3 ($\pm$4.2) |
| FT | 40.3 ($\pm$0.7) | 39.1 ($\pm$1.9) | 39.7 ($\pm$1.3) | 44.4 ($\pm$3.3) | 42.1 ($\pm$4.9) | 43.4 ($\pm$4.1) |
| FT+KL | 39.7 ($\pm$0.6) | 40.2 ($\pm$1.4) | 40.0 ($\pm$1.0) | 45.6 ($\pm$2.2) | 42.8 ($\pm$6.4) | 44.2 ($\pm$4.3) |
| PT | 42.1 ($\pm$0.8) | 40.0 ($\pm$2.6) | 41.1 ($\pm$1.7) | 47.5 ($\pm$2.8) | 43.0 ($\pm$6.1) | 45.3 ($\pm$4.5) |
| **PT+KL** | **42.4** ($\pm$0.4) | **41.8** ($\pm$1.3) | **42.1** ($\pm$0.9) | **47.9** ($\pm$2.1) | **46.6** ($\pm$3.6) | **47.3** ($\pm$2.8) |

Table 10: For the simplicity of the main paper, we report the StD of the results in Table 7

To generate "equivalent" instructions for tasks from seed instructions, we prompted GPT-4 to produce paraphrases of given instructions. We detail our procedure for doing this for each model evaluated below. We used different prompts for these two instruction collections because, in informal development, we found them to produce better paraphrases (respectively). We emphasize that these are instructions from the *training* instruction-tuning set, so this cannot be construed as "cheating", although we acknowledge that there is a small amount of effort necessary to ensure LLMs (GPT-4) produces "good" paraphrases at scale.

**Alpaca**    To generate paraphrases of observed instructions in the Alpaca collection, we sampled 1000 out of 52002 tasks at i.i.d. random and generated instruction paraphrases with GPT-4 via the following prompts.

- "Paraphrase this sentence:\n\n{instruction}Paraphrased sentence:\n\n"
- "Paraphrase this instruction into a longer sentence\n\n\n{instruction}New sentence:\n"
- "You are given an instruction:\n\n{instruction}Now, paraphrase it into a new instruction with equivalent meaning:\n\n"

**Flan**    We first reproduced the held-in instruction-tuning set of Flan-T5 with the pipeline. We randomly sampled 986 data samples from the generated data following the proportion reported in Chung et al. (2022). We generate paraphrases of the selected data with GPT-4 using the following prompts.

- "Here's an input utterance:\n\n{instruction}\n \n \n Now, your task is to paraphrase the input by only changing the instruction but leaving everything else the same.\n Here's the new utterance:\n\n"
- "You are given an utterance which is a combination of task instruction and the actual input. Your job is to paraphrase the task instruction and leave the input unchanged. Here's the utterance to be paraphrased:\n\n\n{instruction}\n\n\n Now, generate the new utterance:\n\n\n"
- "You are provided with the utterance of a specific task and I need you to paraphrase it. The actual input, question, and examples in the task should not be changed. You

> should only paraphrase the instructions. Task:\n\n\n {instruction}\n\n\nThe paraphrased utterance:\n\n\n"

On our informal and subjective assessment, these prompts yielded seemingly good paraphrases; we therefore did not invest time in further "prompt engineering" for this problem, given that our focus in this work is not automated paraphrasing *per se*—rather, we aim only to collect "reasonable" paraphrases that allow us to realize the proposed objective which looks to push representations of semantically similar instructions relatively close together; GPT-4 readily provides outputs servicable in this regard.

### B.3.1 QUALITY STUDY OF INSTRUCTION TUNING DATASETS AND PARAPHRASES

Here we investigate the quality and effectiveness of aligning instruction representations when paraphrases are generated under models aside from GPT-4. We do this to explore the sensitivity of our approach to specific sources of instruction paraphrases, although we emphasize that these might also be manually composed (at an associated cost in effort, of course).

**Training with More Instructions v.s. Inducing Closer Representations** We show that the robustness of instruction-following LLMs could be improved by aligning the representations of semantically equivalent instructions. However, it remains unclear whether this approach is better than training new LLMs from scratch with the extra instructions. Here, we compare the performance and robustness of LLMs re-trained versus LLMs aligned with the new instructions.

| Method | Avg Acc. (MMLU) | Avg Degr. (MMLU) | # Tokens | # Parameters |
|---|---|---|---|---|
| **Flan-T5-XL (3B)** | 47.8 | -0.6 | | |
| - Training w/ new paraphrases | 47.7 | -0.5 | 4.4B | 3B |
| - Aligning w/ new paraphrases | **48.0** | **-0.2** | **390K** | **61K** |
| **Alpaca (7B)** | 40.8 | -2.2 | | |
| - Training w/ new paraphrases | 41.3 | -2.1 | 3.4M | 7B |
| - Aligning w/ new paraphrases | **42.1** | **-0.6** | **233K** | **122K** |

Table 11: Performance on MMLU and BBL when the model is retrained from scratch (with the new instructions) and when the model representation is aligned with the new instructions. # Tokens refers to the total number of tokens used in the process; #Parameters refers to the total number of trainable parameters in the process.

Table 11 shows that even with significantly larger training text and trainable parameters, re-training the model from scratch still underperforms, aligning the representations of equivalent instructions.

**Aligning with Paraphrases from Different Sources** In Section 5, we adopt GPT-4 as the source to generate semantically equivalent instructions. We agree that whether results hold with alternative sources of paraphrases is a worthwhile scientific question. We have replicated the experiment from Section 5 with two other reasonable sources of paraphrases: Llama2-13B Touvron et al. (2023b) and text-DaVinci-003 Ouyang et al. (2022). We reproduce the experiment in Table 7 with Flan-T5-XL.

| | MMLU | | | BBL | | |
|---|---|---|---|---|---|---|
| **Source Model** | OBS. | UNOBS. | Avg. | OBS. | UNOBS. | Avg. |
| Text-DaVinci-003 | 48.1 | 47.7 | 47.9 | 55.4 | 53.0 | 54.2 |
| Llama2-13B | 48.0 | 47.6 | 47.8 | 55.4 | 50.8 | 53.1 |
| **GPT-4** | 48.1 | **47.9** | **48.0** | **55.9** | **53.7** | **54.8** |

Table 12: The performance of the models after aligning representation with soft prefix (**PT+KL** in 7). GPT-4 generated paraphrases improve the performance and robustness of the instruction-following LLMs better than the other two sources.

We take a more in-depth look at the quality of paraphrased instructions generated by GPT-4, Davinci-003, and Llama2-13B. We observed that the performance improvement using the paraphrases as

equivalent instructions is **correlated with the diversity and quality of the paraphrases**. For each source of the paraphrases, we measure the average token length generated, as well as the similarity (ROUGE-2) between the seed instruction and its paraphrases.

Table 13 shows that GPT-4 and DaVinci produce relatively similar paraphrases. The ROUGE-2 score compared with the original instruction is significantly higher than GPT-4 and DaVinci-003, implying a lower diversity and significance in the equivalent instruction set.

| Source Model | Avg. Token Length | Avg. ROUGE-2 w/ seed instruction |
|---|---|---|
| Text-DaVinci-003 | 154 ($\pm 76$) | 47.2 ($\pm 39.5$) |
| Llama2-13B | 286 ($\pm 201$) | 55.3 ($\pm 40.0$) |
| GPT-4 | 165 ($\pm 76$) | 45.8 ($\pm 39.1$) |

Table 13: The statistics of the paraphrased instructions generated from different LLM sources

### B.3.2 Efficacy of Alignment Term

In Table 7, we conduct a detailed ablation study to show the efficacy of the KL term on migitating the robustness issue. Comparing with prefix tuning, the proposed method introduces extra term to cluster the hidden representation produced by the equivalent instructions together, which results in a **1.0** and **3.6** improvement for the unobserved instructions on MMLU and BBL respectively.

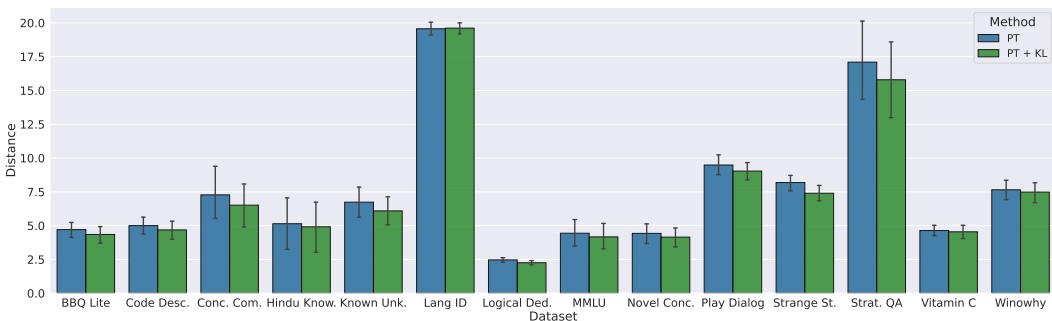

Figure 7: The average $l2$ distance between the unobserved instructions to their closest observed instructions. The result shows that the semantic distance between equivalent instructions is smaller for PT+KL in 13 out of 14 datasets.

### B.4 NLG Experiment Results

We report the full results on four NLG datasets. We use ROUGE-2 (F1) Lin (2004) and BERTScore Zhang et al. (2019) as the automatic metrics for summarization; we use BLEU Papineni et al. (2002) and BERTScore as the automatic metrics for translation. T0++ Sanh et al. (2021) is not evaluated for translation tasks because it is not trained on any non-English tasks.

## C Instruction Robustness with In-context Learning

We have been focused on zero-shot settings, but here we also report results achieved under In-context Learning (ICL). We repeat experiments from Section 4.2 on MMLU with the one-shot and three-shot ICL below. Additionally, we compare the effect of few-shot examples to our proposed representation alignment method:

We observe that while few-shot prompting does indeed mitigate robustness issues to some degree, it does so with high variance, compared to the proposed alignment technique. Furthermore, the selection of in-context examples also affects the performance and robustness of k-shot prompting. We repeat the Alpaca-7B (Three-shots) experiment 10 more times with random examples sampled from the test set and the standard deviation raises from **3.9** to **5.8**, indicating the instability of the

| Model | CNN/DM $R_2$ Avg. (Std.) | CNN/DM BERT Avg. (Std.) | XSUM $R_2$ Avg. (Std.) | XSUM BERT Avg. (Std.) | MEDIASUM $R_2$ Avg. (Std.) | MEDIASUM BERT Avg. (Std.) | WMT19 FR-DE $B_2$ Avg. (Std.) | WMT19 FR-DE BERT Avg. (Std.) |
|---|---|---|---|---|---|---|---|---|
| **Flan-T5-Small** | | | | | | | | |
| OBSERVED | **32.0** (±6.1) | **95.2** (±0.2) | **24.6** (±0.3) | **96.1** (±0.0) | 16.5 (±1.4) | **95.9** (±0.1) | **4.6** (±0.5) | **94.2** (±0.2) |
| UNOBSERVED | 31.5 (±5.6) | 95.1 (±0.2) | 23.6 (±1.3) | 96.0 (±0.0) | **18.6** (±2.3) | 95.9 (±0.1) | 3.8 (±1.5) | 93.6 (±1.0) |
| **Performance Δ** | ↓ 0.5 | ↓ 0.1 | ↓ 0.9 | ↓ 0.0 | ↑ 2.1 | ↓ 0.0 | ↓ 0.8 | ↓ 0.6 |
| **Flan-T5-Base** | | | | | | | | |
| OBSERVED | 27.7 (±0.3) | **95.1** (±0.1) | **29.4** (±0.2) | **96.0** (±0.0) | 15.3 (±0.8) | **95.8** (±0.1) | **13.5** (±1.0) | **94.7** (±0.0) |
| UNOBSERVED | **28.7** (±0.6) | 95.0 (±0.0) | 29.1 (±0.5) | 96.0 (±0.0) | **17.1** (±4.5) | 95.8 (±0.1) | 11.8 (±3.9) | 94.5 (±0.4) |
| **Performance Δ** | ↑ 1.0 | ↓ 0.1 | ↓ 0.4 | ↓ 0.0 | ↑ 1.7 | ↓ 0.0 | ↓ 1.7 | ↓ 0.2 |
| **Flan-T5-Large** | | | | | | | | |
| OBSERVED | **39.6** (±1.0) | **95.4** (±0.2) | **32.9** (±1.1) | **95.9** (±0.1) | **20.5** (±2.5) | **95.9** (±0.2) | **15.9** (±2.3) | **94.5** (±0.1) |
| UNOBSERVED | 34.3 (±3.9) | 95.2 (±0.3) | 32.1 (±1.4) | **95.9** (±0.1) | 18.7 (±1.5) | 95.9 (±0.1) | 15.2 (±5.4) | 94.5 (±0.3) |
| **Performance Δ** | ↓ 5.3 | ↓ 0.2 | ↓ 0.9 | ↓ 0.0 | ↓ 1.8 | ↓ 0.1 | ↓ 0.7 | ↓ 0.0 |
| **Flan-T5-3B** | | | | | | | | |
| OBSERVED | **41.5** (±0.1) | **95.3** (±0.0) | **36.1** (±1.0) | **95.9** (±0.0) | **23.4** (±6.2) | **96.0** (±0.1) | **19.2** (±1.8) | **94.5** (±0.1) |
| UNOBSERVED | 32.3 (±2.4) | 95.3 (±0.1) | 35.7 (±1.1) | 95.9 (±0.0) | 18.0 (±1.4) | 95.7 (±0.1) | 18.4 (±3.0) | 94.5 (±0.1) |
| **Performance Δ** | ↓ 9.1 | ↓ 0.0 | ↓ 0.4 | ↓ 0.0 | ↓ 5.4 | ↓ 0.3 | ↓ 0.7 | ↓ 0.0 |
| **Alpaca-7B** | | | | | | | | |
| OBSERVED | **19.2** (±3.4) | 93.0 (±0.6) | **12.4** (±1.3) | 92.3 (±0.2) | **9.9** (±1.6) | **92.9** (±0.5) | **16.5** (±6.8) | **94.1** (±0.6) |
| UNOBSERVED | 16.8 (±0.9) | **93.3** (±0.5) | 8.3 (±0.4) | **93.3** (±0.9) | 9.1 (±0.7) | 92.9 (±0.6) | 16.2 (±6.9) | 93.7 (±1.0) |
| **Performance Δ** | ↓ 2.4 | ↑ 0.3 | ↓ 4.1 | ↑ 1.0 | ↓ 0.7 | ↓ 0.0 | ↓ 0.3 | ↓ 0.3 |
| **Flan-T5-11B** | | | | | | | | |
| OBSERVED | **33.6** (±0.2) | **95.1** (±0.0) | **39.9** (±0.1) | **95.9** (±0.0) | **22.2** (±2.3) | 95.7 (±0.1) | **22.2** (±0.4) | **94.6** (±0.0) |
| UNOBSERVED | 32.0 (±2.3) | 95.1 (±0.1) | 39.7 (±0.3) | 95.9 (±0.0) | 18.0 (±2.2) | **95.8** (±0.1) | 22.0 (±1.3) | 94.6 (±0.0) |
| **Performance Δ** | ↓ 1.6 | ↓ 0.0 | ↓ 0.3 | ↓ 0.0 | ↓ 4.2 | ↑ 0.2 | ↓ 0.2 | ↓ 0.0 |
| **T0-11B** | | | | | | | | |
| OBSERVED | **35.1** (±0.1) | **95.3** (±0.0) | **40.2** (±0.3) | **95.9** (±0.0) | **18.9** (±4.8) | **95.6** (±0.2) | - | - |
| UNOBSERVED | 35.0 (±0.2) | 95.3 (±0.0) | 40.1 (±0.5) | 95.9 (±0.0) | 16.5 (±0.8) | 95.4 (±0.1) | - | - |
| **Performance Δ** | ↓ 0.1 | ↓ 0.0 | ↓ 0.1 | ↓ 0.0 | ↓ 2.4 | ↓ 0.1 | - | - |
| **Alpaca-13B** | | | | | | | | |
| OBSERVED | **19.3** (±3.0) | **94.3** (±0.3) | **10.3** (±3.0) | **94.9** (±0.2) | **7.1** (±0.3) | 94.2 (±0.3) | 14.0 (±5.8) | **94.8** (±0.0) |
| UNOBSERVED | 16.1 (±0.9) | 94.1 (±0.1) | 8.0 (±0.5) | 94.8 (±0.1) | 6.9 (±0.3) | **94.5** (±0.3) | **15.4** (±6.1) | 94.8 (±0.0) |
| **Performance Δ** | ↓ 3.3 | ↓ 0.2 | ↓ 2.3 | ↓ 0.1 | ↓ 0.2 | ↑ 0.2 | ↑ 1.4 | ↓ 0.0 |

Table 14: Full results using observed and unobserved instructions across NLG tasks.

| Models | Avg. Acc. (Performance) | (Obs - Unobs) Acc. (Robustness) |
|---|---|---|
| **Flan-T5-XL** | | |
| - 0-shot | 47.8 (±0.6) | 0.6 |
| - 1-shot | 47.5 (±0.9) | 0.2 |
| - 3-shot | **48.1** (±1.9) | **0.2** |
| - 0-shot + Aligned | 48.0 (±0.3) | **0.2** |
| **Alpaca-7B** | | |
| - 0-shot | 40.8 (±1.4) | 2.2 |
| - 1-shot | 41.3 (±2.7) | 1.1 |
| - 3-shot | 41.7 (±3.9) | 1.4 |
| - 0-shot + Aligned | **42.1** (±0.3) | **0.6** |

Table 15: The ICL experiments on MMLU with one-shot and three-shot examples provided.

method. With the above results, we argue that the effect of in-context learning on robustness is a hybrid research question, in which the effect of examples needs to be studied independently.

We provide the disaggregated results for few-shot experiments on MMLU and BBL in Appendix K.2.

# D LLAMA2 V.S. LLAMA: DOES BETTER PRE-TRAINING PROVIDE STRONGER INSTRUCTION-FOLLOWING ROBUSTNESS?

To provide a better understanding of what factors (aside from instruction-tuning) correlate with the ability of LLMs to robustly follow instructions, we compare the performance and brittleness of Llama2 Touvron et al. (2023b) and Llama Touvron et al. (2023a) when instruction-tuned with Alpaca Taori et al. (2023). Llama2 Touvron et al. (2023b) is an LLM trained on a relatively new mix of publicly available data. This pre-training corpus is about $40\%$ larger than that used to train the original Llama.

## D.1 TRAINING DETAILS

We trained both Llama-7B and Llama2-7B on 4 A100 80GB GPUs with the Fully Sharded Data Parallel (FSDP) strategy. We adopted the exact same hyperparameters for training: We use the AdamW optimizer Loshchilov & Hutter (2017) with a learning rate of $2 \times 10^{-5}$. We use a cosine learning rate scheduler with a warmup ratio of 0.03. We trained for a total of 3 epochs over 52k instruction and answer pairs collected in Alpaca, with a batch size of 128. Llama2 and Llama have different max context window sizes; to simplify things we set both to use a fixed max length of 512.

## D.2 EVALUATION DETAILS

Following the same procedure in Section 4.2, we reproduce experiments to evaluate the performance of Llama2-Alpaca on observed and unobserved instructions. We omit two held-in datasets that we evaluated to avoid data leakage. For each category of tasks (MMLU, QA, MC, BC, Sum, Trans), the average performance is calculated, as well as the percentage delta of the performance between observed and unobserved instructions on each task.

## D.3 RESULTS

Figure 8 shows that although Llama2-Alpaca is consistently superior across all categories of tasks compared to Llama-Alpaca, the robustness (performance difference between observed and unobserved instructions) of the model **does not** always improve. On summarization datasets, binary classification, and MMLU, the robustness of Llama2-Alpaca is better than Llama-Alpaca, whereas, on translation datasets, QA, and multi-class classification, the instruction-following robustness is worse. Hence, it cannot be concluded that better pre-training provides stronger instruction-following robustness after instruction-tuning.

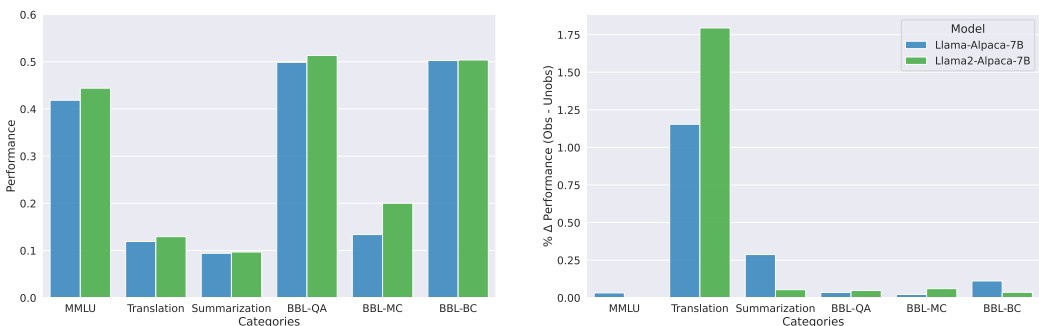

Figure 8: The overall results of Llama-Alpaca v.s. Llama2-Alpaca on the six aforementioned categories of held-out datasets. It could be seen that although the overall average performance of Llama2-Alpaca is slightly better than Llama-Alpaca (Left), there is no clear pattern that the performance difference between observed and unobserved instructions are narrowed down (Right).

# E DIFFERENT (SIMILAR) OBJECTIVES INDUCE SIMILAR REPRESENTATIONS

Besides the KL objective reported in the main paper, we explored two other objectives that similarly aim to align representations for practically equivalent instructions. Our goal with these experiments is to establish that the primary mechanism to improve robustness has to do with encouraging the model to internally represent equivalent instructions similarly; the particular choice of objective for realizing this is not critical.

## E.1 CONTRASTIVE LEARNING ON PENULTIMATE LAYER LOGITS

We use an extra contrastive term to push the inequivalent instructions away while aligning the equivalent instructions. Following the same batching strategy in Section 5, one training batch contains 2 seed task instructions and $n$ paraphrases of each. While all the other terms and loss remain the same as 5, we introduce the contrastive KL loss $\mathcal{L}_{KL}$ as:

$$\mathcal{L}_{KL} = \frac{1}{N-1} \sum_{i \neq r, r'}^{N} \sum_{j} [\text{KL}(\hat{y}_i^{(j)} || \hat{y}_r^{(j)}) - \text{KL}(\hat{y}_i^{(j)} || \hat{y}_{r'}^{(j)})]$$

Where for vocabulary distribution of each paraphrase $\hat{y}_i$, $\hat{y}_r$ is the seed instruction it is equivalent to, and $\hat{y_{r'}}$ is the seed instruction it isn't equivalent to.

## E.2 MSE LOSS ON THE PREFIX ENCODER HIDDEN STATE

Instead of having to optimize the logits to induce closer representations for equivalent instructions, we tried to directly align the hidden representation of the soft prefix from the last encoder layer we they are concatenated to equivalent instructions. With the same batching strategy in Section 5, we optimize:

$$\underset{\boldsymbol{\theta}^{d \times n}}{\text{argmax}} \frac{1}{N-1} \sum_{i \neq r}^{N} \mathcal{L}_{\text{MSE}}(\hat{h}_i, \hat{h}_r)$$

Where $\hat{h}_i$ and $\hat{h}_r$ are given by:

$$\hat{h}_i = \text{Encoder}([\boldsymbol{\theta} : \text{Emb}(x_i)])$$

This method is effective and fast but cannot be used with decoder-only architecture LLMs.

## E.3 RESULTS COMPARISON

In some sense all of these objectives are equivalent in that we are trying to induce closer representations for similar instructions. The experimental results are reproduced here (we omit variances for simplicity):

| Methods | Avg. Obs. Acc. (MMLU) | Avg. Unobs. Acc. (MMLU) |
|---|---|---|
| FT + KL | 41.8 | 43.6 |
| PT + KL | **48.1** | 47.9 |
| FT + Contrastive KL | 41.9 | 43.6 |
| PT + Contrastive KL | 47.9 | 47.7 |
| MSE on EncoderHS | 48.1 | **48.0** |

Table 16: The results of the representation alignment experiment on Flan-T5-3B with two alternative objectives that incent closer distance between similar instructions.

Using these objectives to attempt to align instructions performs comparably (in terms of robustness) to the proposed soft prompt alignment strategy, but they are slower to converge than this method. Furthermore, the second approach is only suitable for encoder-decoder LLMs like Flan-T5.

# F "CLOSER LOOK" AT SEMANTIC DISTANCES

In Section 4.5 we reported a rather peculiar result: Over all the tasks we evaluated, incorrect but observed instructions perform better on average than correct but unobserved instructions. Does our proposed method for improving robustness (by aligning representations in latent space) change this particular result? We re-run the experiment reported in Figure 3 again with a model fit using the objective proposed in Section 5.

In Table 17, we report a non-trivial improvement in all settings. Interestingly, with the proposed method (**PT+KL**) the average performance when using instructions that are **observed** and incorrect (but which belong to the same genre of task) still performs better than the task designer result; that said, the gap in performance is slightly narrowed compared to the baseline (7.1 difference for PT + KL vs 8).

## F.1 NEGATION

Previous studies Webson & Pavlick (2022); Jang et al. (2023) suggest the insensitivity to negation as an indicator of unrobust instruction-following ability of LLMs. The results reported in Table 17 and Section 4.5 also support such finding. Additionally, Table 17 shows that despite generally improving model's performance and robustness in other settings, the proposed method has failed to bring significant improvement (drop in accuracy) in the **Negated** setting. Hence, we slight tilt the objective to see if training on "pushing opposite instructions away" would result in a more robust instruction-following ability for recognizing negation.

Similar to Section 5, we take the same 1k seed instances from the flan collection and prompt (with 3 prompts) the GPT-4 to rewrite them with completely opposite instruction. All the other parameters for generation stay the same.

Naturally, instead of clustering equivalent instructions together, we want to push the negated instructions away from the original instruction:

$$\mathcal{L} = (1 + \lambda)\mathcal{L}_{\text{CE}} - \lambda\mathcal{L}_{\text{KL}}, \quad \mathcal{L}_{\text{KL}} = \frac{1}{N-1}\sum_{i \neq r}^{N}\sum_{j}(-1)\text{KL}(\hat{y}_i^{(j)}||\hat{y}_r^{(j)})$$

We train the model with the exact same hyperparameters in Section 5 for PT + KL and run it on the same benchmarks with the six different settings. We report the results along with the original method, denoted as Negation[†]. Additionlly, We also report the results of jointly optimizing with negated and paraphrased instructions, denoted as Joint.

| Model | Closest | Incorrect | Negated | Task Designer | Collected | Nonsensical |
|---|---|---|---|---|---|---|
| FLAN-T5-XL | | | | | | |
| - Baseline | 58.1 ±1.8 | 52.2 ±8.6 | 48.3 ±3.8 | 44.2 | 55.3 ±4.5 | 30.6 ±4.4 |
| - PT + KL | **60.5** ±2.4 | **54.2** ±7.2 | 47.1 ±4.9 | **47.1** | **56.7** ±3.9 | 38.0 ±4.1 |
| - Negation[†] | 58.6 ±2.0 | 52.7 ±9.2 | **43.0** ±4.9 | 46.5 | 55.6 ±4.3 | 35.7 ±5.6 |
| - Joint | 59.3 ±1.7 | 52.8 ±8.7 | 44.2 ±4.5 | 46.6 | 56.1 ±3.5 | 36.1 ±5.1 |

Table 17: We report average result of six settings over benchmarks. For Negated setting, a negation is applied to the instruction so a lower accuracy is desired. Despite the fact that in Negation[†] setting the model performs comparably with the baseline, its ability to recognize negation in the instruction is significantly improved. Training with the joint loss does realize some of the advantages of both negative and paraphrase alignment methods. Compared with the standard fine-tuned baseline (no KL), we see consistent improvements across all settings. But we see stronger improvements in the respective settings when we optimize for only paraphrases or negations alone.

## G  CATASTROPHIC FORGETTING

In Table 7 one can observe that simply fine-tuning (FT) on the augmented paraphrase data significantly degredes performance when using both observed and unobserved instructions. We hypothesize that this may be an instance of *catastrophic forgetting*, where when we fine-tune the instruction-tuned model in this way (on a relatively small set of paraphrased instruction instances) it compromises its ability to follow instructions more general.

We design a simple study to test this hypothesis. Following the same setting of **FT** in Section 5, we finetune FLAN-T5-XL (which is instruction tuned) and T5-XL (which is not) on the same augmented data (4k paraphrases of task instructions) for 25 epochs, and evaluate the performance on (A) the validation set sampled from the flan collection Chung et al. (2022), (B) MMLU with observed instructions, and (C) MMLU with unobserved instructions.

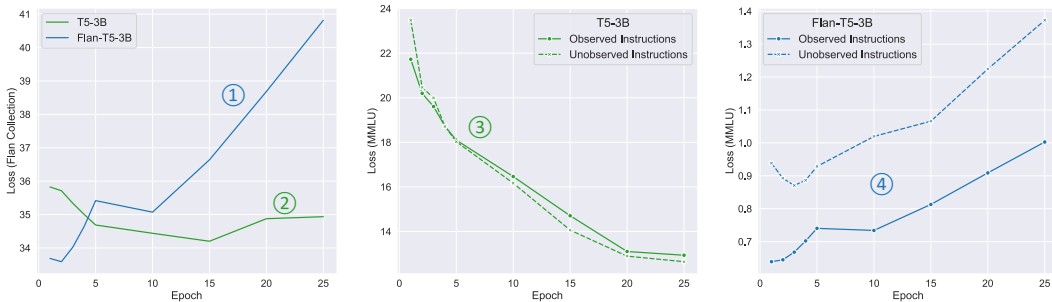

Figure 9: Loss as a function of training epoch for FLAN-T5-3B and T5-3B, fine-tuned on the paraphrased instruction set. The first (left) plot shows the validation loss on the flan collection for both models; the second (middle) plot shows the validation loss on MMLU for T5-3B; the third (right) plot shows the validation loss on MMU for FLAN-T5-3B. The loss for FLAN-T5-3B increases almost monotonically (Line 1 and Lines 4); this strongly suggests catastrophic forgetting. Coult it be that the collected paraphrased instructions are simply not informative? The decreasing loss for T5-3B — indicated by Line 2 and Lines 3 — suggests otherwise, i.e., that the data does contain signal for instruction tuning (on a base model), but the set is somewhat small and degrades performance when the Flan checkpoint is used for initialization.

What is happening at the representational level in these cases? In Figure 10 we compare the average similarity (analogous to Table 5) between observed and unobserved instructions after 25 epochs of fine-tuning. Rather than closing the "semantic gap", fine-tuning FLAN-T5-XL (without the proposed KL term) *increases* it.

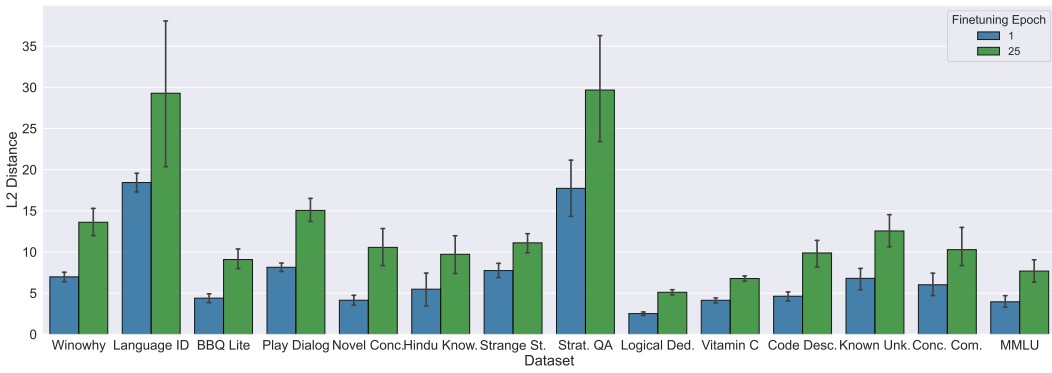

Figure 10: The average $\ell 2$ Distance between the unobserved instructions to their closest observed instructions on each datasets at epoch 1 and epoch 25. This result indicates that simply fine-tuning the model with paraphrases **does not** help to align equivalent instructions.

## H  CLOSER LOOK TO ROBUSTNESS AS FUNCTION OF SCALE?

While we maintain our stance that the success of instruction-tuning for boosting the performance of modestly-sized LLMs is the most exciting aspect of this method, and therefore worthy on its own of study, we nonetheless agree that studying the relationship between robustness and scale of such models is interesting.

However, we can draw valid comparisons only if we keep all other variables (model architecture, pretraining data, and instruction tuning set) fixed. Therefore, we are not able to study black-boxed models (like GPT-3.5 or GPT-4) using this approach. To include additional scale in our work, we therefore conduct the full experiments on different sizes of Flan-T5 ranging from 70M to 11B 4.3, as well as Flan-UL2 (20B), and we report the change of **delta performance** (observed instructions performance - unobserved instructions performance) in the table below:

| Model | Number of Parameters | MMLU | QA | MC | BC |
|---|---|---|---|---|---|
| Flan-T5-Small | 70M | -2.6 ± 8.3 | 4.1 ± 3.1 | -0.4 ± 1.4 | -7.9 ± 12.0 |
| Flan-T5-Base | 250M | 0.0 ± 0.9 | 1.6 ± 6.2 | 1.2 ± 4.3 | 1.5 ± 8.3 |
| Flan-T5-Large | 770M | 0.1 ± 0.5 | 3.1 ± 6.4 | 11.4 ± 2.1 | 1.6 ± 6.1 |
| Flan-T5-XL | 3B | 0.6 ± 0.7 | 3.0 ± 4.6 | 3.5 ± 3.2 | 5.5 ± 7.8 |
| Flan-T5-XXL | 11B | 0.5 ± 0.6 | 3.4 ± 5.7 | 2.8 ± 3.1 | 2.0 ± 9.1 |
| Flan-T5-UL2 | 20B | 0.2 ± 0.5 | 3.4 ± 7.3 | 0.6 ± 0.3 | 1.9 ± 7.1 |

Table 18: The aggregated result of the delta performance (bserved instructions performance - unobserved instructions performance) as the Flan-T5 model scales. The results above (aggregated by task category) show that there is no apparent relationship between model size and the delta performance (our measure of robustness).

More scientifically, we perform linear regression on the size of the model v.s. (observed - unobserved performance) over data points from all datasets, this yields a slope of $4.45 \times 10^{-13}$ with a p-value of 0.087 ($R^2$ is 0.002). This suggests that model size may weakly correlate with increased robustness, but this relationship is not significant the $p = 0.05$ level, and the relationship is tenuous. In other words, scale does seem to improve robustness, but only very moderately.

# I REPRESENTATIONAL SIMILARITY AND MODEL PERFORMANCE

We re-generate the visualization from Figure 4 for all T5 model sizes in Figure . The qualitative result—representations of tokens following observed instructions are generally dissimilar from those following unobserved instructions—remains largely consistent, although it is less pronounced, e.g., for XL (in particular here, the MMLU samples are not as entangled as we might expect).

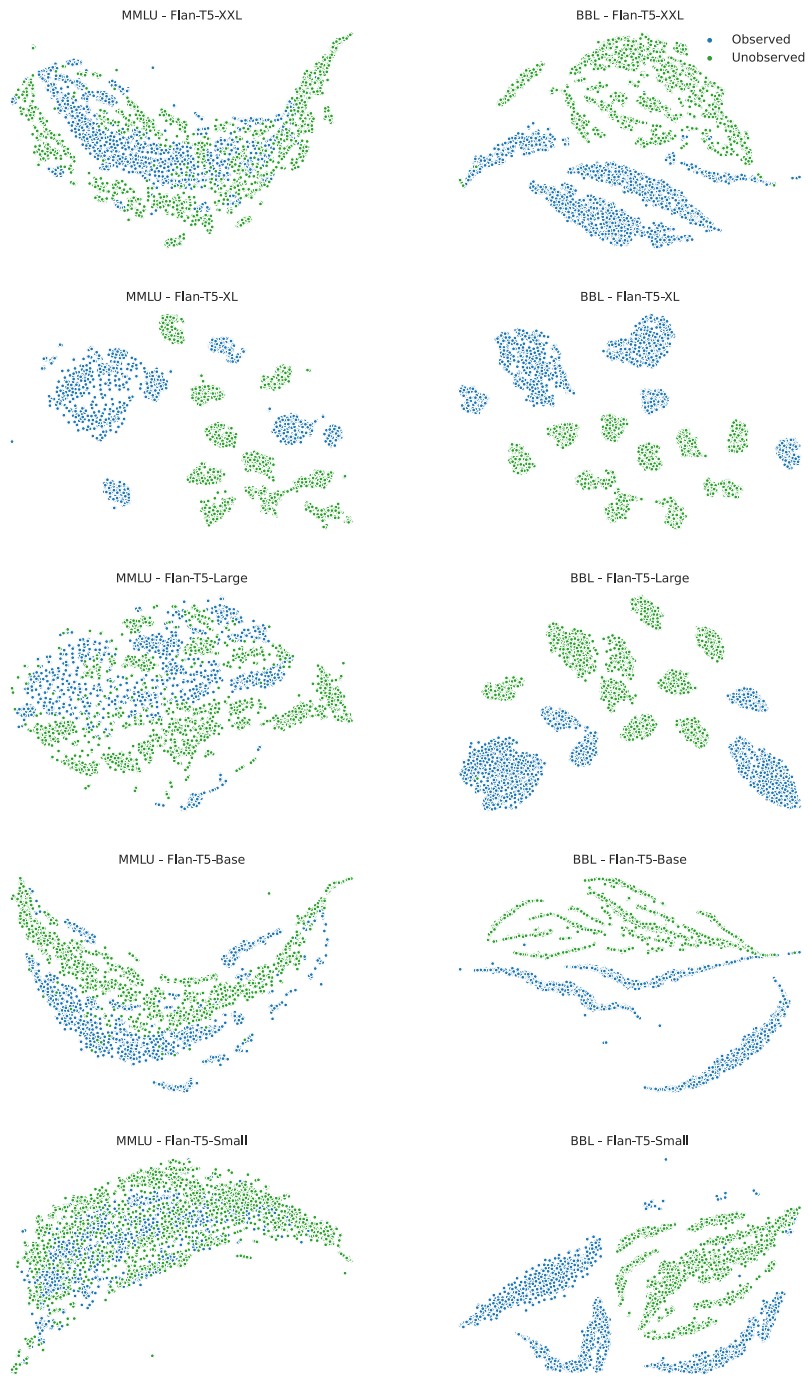

Figure 11: We reproduce Figure 4 from the main paper over all T5 model sizes.

## J COLLECTION PROCEDURES AND SURVEYS

We recruited in total 36 annotators to contribute diverse human-written instructions for various NLP tasks. These were graduate students in NLP who volunteered to write instructions following an open request. We explicitly asked volunteers not to spend more than 30 minutes on this task. We estimate that the maximum time spent by any one individual on this task was one hour, and most were (much) quicker. In light of the minimal workload per individual being done on a strictly volunteer basis, we did not offer compensationl; we emphasize that it was made clear upfront to volunteers that this quick amount of work would be uncompensated.

The entire annotation process was done online (via a shared Google Drive directory) with detailed instructions provided. We reproduce the template of the invitation letter, annotation instructions, and annotation form below:

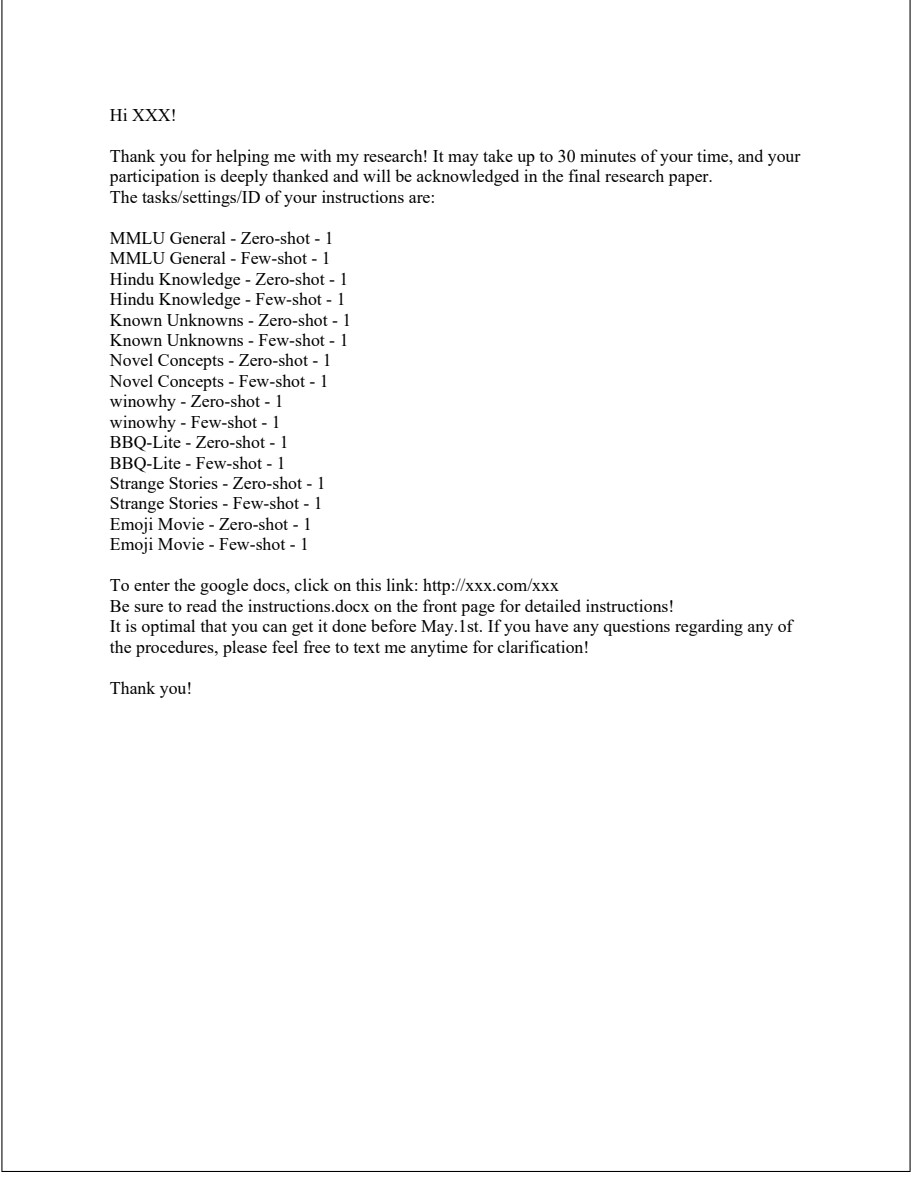

Figure 12: Invitation note sent to participant

First, I would like to express my appreciation for helping with my research project again!

**Background**
This research aims to evaluate the robustness of the instruction-tuned Language Models (LMs) with respect to the variation of instructions in zero-shot or few-shot settings. It is commonly acknowledged that multitask instruction tuning on a language model improves its zero-shot and few-shot ability. The model can understand and generalize to unseen instructions that users provide at inference time.

For instance, I use this instruction (prompt) as the Prefix:

"Complete this code written in Java SE11 …"

to the actual code, I want to complete, the LM can understand the task and perform inference accordingly.

**Goal**
As an NLP practitioner and expert, you can provide **instructions** that will prompt the instruction-tuned LMs well for **the given tasks**. The models in which the instructions might be evaluated are:

- GPT-4 / ChatGPT
- Text Davinci
- Flan-PaLM
- Flan-T5
- T0++
- mT0
- MetaICL
- OPT-IML
- ChatGLM
- Alpaca

You are very well come to use your experience on these models to come up with the instruction you think will **perform the best**.

**Tasks**
The participation will take approximately **30 minutes**. You will be given **10-15** tasks/settings. For each task/setting, you are going to put your instruction in the row indicated by the order number. For instance:

"Auto Debugging - Zero-Shot - 5"

means that you are assigned to write an instruction on the task **"Auto Debugging"** with the setting **"Zero-shot,"** and you are putting your answer in the row with ID **5**.

Figure 13: The first page of the instruction given to the annotator

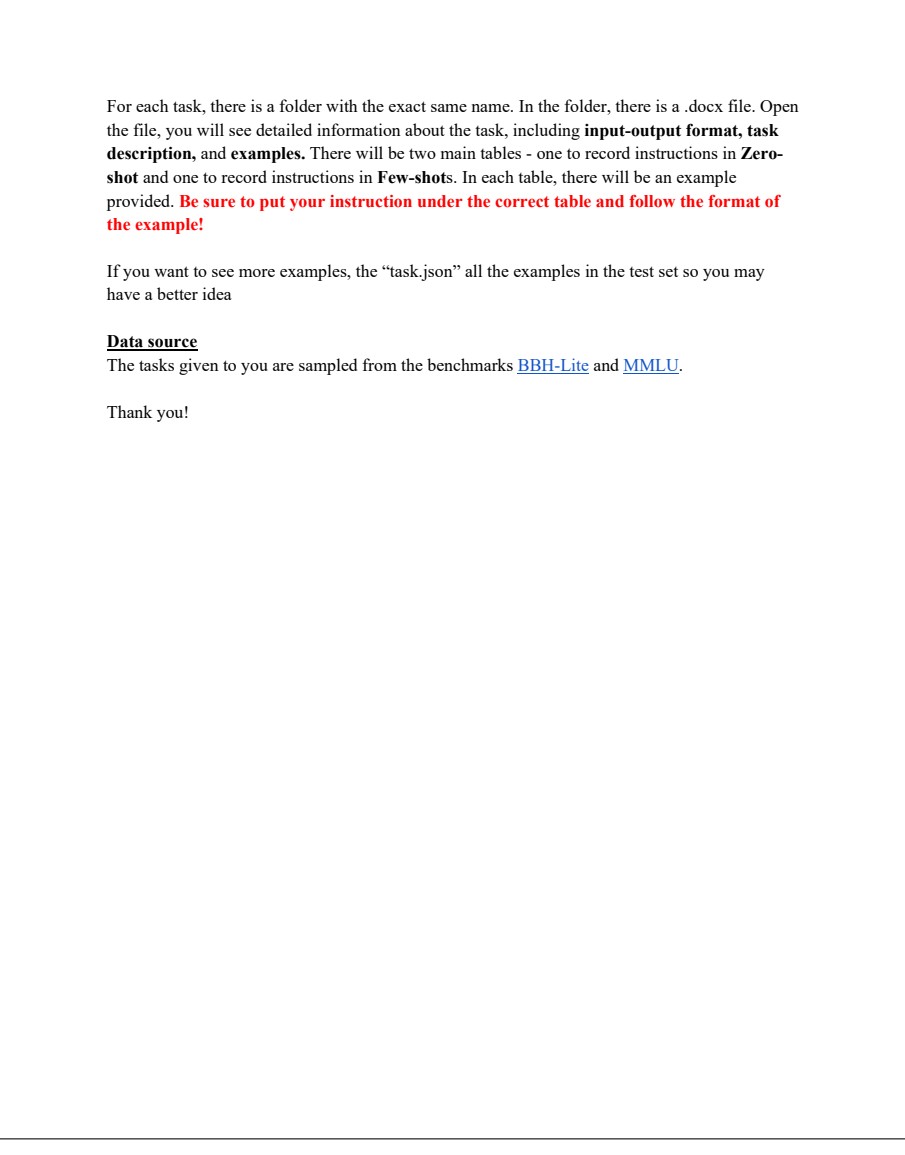

For each task, there is a folder with the exact same name. In the folder, there is a .docx file. Open the file, you will see detailed information about the task, including **input-output format, task description,** and **examples.** There will be two main tables - one to record instructions in **Zero-shot** and one to record instructions in **Few-shot**s. In each table, there will be an example provided. **Be sure to put your instruction under the correct table and follow the format of the example!**

If you want to see more examples, the "task.json" all the examples in the test set so you may have a better idea

**Data source**
The tasks given to you are sampled from the benchmarks BBH-Lite and MMLU.

Thank you!

Figure 14: The second page of the instruction given to the annotator

Thank you for helping me on this research project! The goal is to gather instructions from experienced **NLP researchers** on various downstream tasks incorporated in the benchmark *BBH*. Your task is to:

- Write down the instruction (prompt) for this task that you think will **work the best** for this task on **instruction-tuned** Seq2Seq LMs (Flan-T5-XXL, Davinci-text-003, OPT-IML, etc.) at zero-shot and few-shots (in-context learning).

- Please put your instruction in the **corresponding row** in the tables. The few-shots table is one page below the zero-shot table. Please use {…} to denote corresponding information. Note: you **do not** need to use all the information if you think some are distractions.

- For multiple choice tasks, you may either formulate the instruction to let the model output the **exact text** or **number/letter** of the text. Same goes for classification task.

- **Task Information** provides an overview of the task, including its input, output, and task description; **Example** provides an example to the test set so you may have a better grasp of the nature of the task; the tables of **Zero-shot Instruction** and **Few-shots Instruction** are in the following pages.

- Instead of using "\n" or "\t", you may directly use enter or tab.

- The given example also represents the **average** length of the input/output for this task. You may assume the maximum token length of the LM is **4096**

**Task Information**

| Dataset | BIG-Bench |
|---|---|
| Task | Code Line Description |
| Metric | Accuracy |
| Task description | Give an English language description of Python code |
| Input | program, choiceA, choiceB, choiceC, choiceD |
| Output | answer |

Figure 15: The first page of the dataset information

**Example:**

Input
- **program:** for i in range(23):\n\t print(i)
- **choiceA:** prints values from 0 to 22
- **choiceB:** computes first 10 prime numbers
- **choiceC:** prints values from 1 to 10
- **choiceD:** prints 'hello world' to the terminal

Output
- **answer:** prints values from 0 to 22 / **A**

**Zero-shot Instruction:**

You are given:
- {program}: the text sequence of the input code
- {choiceA}, {choiceB}, {choiceC}, {choiceD}: choices of the interpretation

| ID | Instruction |
|---|---|
| Example | Give an English language description of Python code{program} A. {choiceA} B. {choiceB} C. {choiceC} D. {choiceD}

English language description: |
| 1 | |
| 2 | |
| 3 | |
| 4 | |
| 5 | |
| 6 | |
| 7 | |
| 8 | |
| 9 | |
| 10 | |

Figure 16: The second page of the dataset information

# K   DISAGGREGATED RESULTS

## K.1   MAIN RESULTS AND SCALING RESULTS

In the main paper we reported aggregated results over benchmark corpora. Here we report results on individual datasets for BBL. For MMLU, we evaluate all 57 datasets together, because these are all QA tasks (and we would want a QA model to be capable of answering questions across a diverse set of domains). We report means and stadard deviations of the accuracies achieved over all instructions in Table 20. The numbers on the left of the setting suggest the number of instructions used. We also share even more granular results—reporting the performance for each instruction—in CSV files provided in the supplemental material.

| MMLU | | | | | |
|---|---|---|---|---|---|
| **Model** | Flan-T5-XL | Flan-T5-XXL | T0pp-11B | Alpaca-7B | Alpaca-13B |
| **MMLU** | | | | | |
| OBSERVED | **48.1 ($\pm$ 0.3)** | **53.2 ($\pm$ 0.2)** | 48.3 ($\pm$ 0.9) | **41.9 ($\pm$ 0.6)** | **47.8 ($\pm$ 0.5)** |
| UNOBSERVED | 47.5 ($\pm$ 0.9) | 52.7 ($\pm$ 0.8) | **48.5 ($\pm$ 0.9)** | 39.7 ($\pm$ 2.2) | 47.0 ($\pm$ 0.8) |

Table 19: Granular results corresponding to the subset reported in Table 3, on each dataset of MMLU. We treat all tasks in MMLU equally as general QA and computed the overall accuracy.

| MMLU | | | | | |
|---|---|---|---|---|---|
| **Size Variance** | Small (80M) | Base (250M) | Large (780M) | XL (3B) | XXL (11B) |
| **MMLU** | | | | | |
| OBSERVED | 29.4 ($\pm$ 1.0) | **34.1 ($\pm$ 0.4)** | **41.1 ($\pm$ 0.2)** | **48.1 ($\pm$ 0.3)** | **53.2 ($\pm$ 0.2)** |
| UNOBSERVED | **29.6 ($\pm$ 0.9)** | 33.8 ($\pm$ 1.2) | 40.7 ($\pm$ 0.7) | 47.5 ($\pm$ 0.9) | 52.7 ($\pm$ 0.8) |

Table 20: Granular results corresponding to the subset reported in Figure 2b, on each dataset of MMLU. We treat all tasks in MMLU equally as general QA and computed the overall accuracy.

| | BBL-QA | | | | |
|---|---|---|---|---|---|
| **Model** | Flan-T5-XL | Flan-T5-XXL | T0pp-11B | Alpaca-7B | Alpaca-13B |
| **BBQ Lite** | | | | | |
| OBSERVED | 66.5 (± 1.5) | **77.4 (± 2.4)** | **51.8 (± 5.3)** | 32.6 (± 1.0) | 43.5 (± 1.4) |
| UNOBSERVED | **67.0 (± 7.0)** | 73.7 (± 11.4) | 51.6 (± 3.0) | **33.1 (± 1.3)** | **45.5 (± 2.9)** |
| **Code Desc.** | | | | | |
| OBSERVED | **73.6 (± 3.4)** | **83.6 (± 1.7)** | 70.3 (± 3.0) | **70.2 (± 2.5)** | **85.2 (± 2.4)** |
| UNOBSERVED | 69.7 (± 12.4) | 72.9 (± 22.2) | **70.5 (± 3.7)** | 67.5 (± 11.3) | 82.2 (± 8.5) |
| **Hindu Know.** | | | | | |
| OBSERVED | **52.4 (± 1.6)** | 53.9 (± 1.8) | **57.1 (± 2.5)** | 50.9 (± 2.1) | 63.8 (± 0.7) |
| UNOBSERVED | 47.1 (± 5.4) | **56.5 (± 3.5)** | 53.2 (± 3.0) | 49.8 (± 5.1) | **63.9 (± 1.1)** |
| **Known Unk.** | | | | | |
| OBSERVED | **79.3 (± 2.5)** | **84.7 (± 2.1)** | 70.9 (± 10.2) | **75.2 (± 4.7)** | **81.9 (± 4.3)** |
| UNOBSERVED | 69.0 (± 6.7) | 80.6 (± 8.1) | **76.1 (± 5.9)** | 60.9 (± 11.2) | 71.1 (± 16.3) |
| **Logical Ded.** | | | | | |
| OBSERVED | **52.5 (± 1.0)** | **58.0 (± 0.7)** | **45.5 (± 0.8)** | **25.5 (± 1.1)** | **29.2 (± 1.3)** |
| UNOBSERVED | 52.1 (± 1.1) | 57.8 (± 0.6) | 45.3 (± 1.2) | 24.5 (± 2.3) | 28.0 (± 1.6) |
| **Novel Conc.** | | | | | |
| OBSERVED | 29.8 (± 2.4) | **50.1 (± 1.9)** | 28.8 (± 2.9) | **37.2 (± 5.2)** | **20.0 (± 3.1)** |
| UNOBSERVED | **31.2 (± 5.0)** | 46.0 (± 5.4) | **31.5 (± 5.1)** | 36.1 (± 7.6) | 19.6 (± 4.0) |
| **Logic Grid** | | | | | |
| OBSERVED | **41.8 (± 1.1)** | **43.2 (± 1.8)** | **37.6 (± 1.7)** | 24.4 (± 1.9) | **29.3 (± 0.8)** |
| UNOBSERVED | 38.6 (± 5.4) | 39.6 (± 4.4) | 36.4 (± 3.6) | **25.4 (± 1.1)** | 28.7 (± 1.2) |
| **Conc. Com.** | | | | | |
| OBSERVED | **75.9 (± 1.8)** | **75.0 (± 2.6)** | 73.3 (± 2.3) | **58.7 (± 4.0)** | **63.0 (± 2.2)** |
| UNOBSERVED | 75.0 (± 1.9) | 73.6 (± 4.8) | **74.2 (± 2.6)** | 55.9 (± 6.2) | 61.1 (± 3.3) |

Table 21: Granular results corresponding to the subset reported in Table 3, on each dataset of category BBL-QA.

| BBL-QA | | | | | |
|---|---|---|---|---|---|
| **Size Variance** | Small (80M) | Base (250M) | Large (780M) | XL (3B) | XXL (11B) |
| **BBQ Lite** | | | | | |
| Observed | 28.3 (± 1.3) | **51.5 (± 1.4)** | 56.6 (± 2.0) | 66.5 (± 1.5) | **77.4 (± 2.4)** |
| Unobserved | **28.6 (± 4.3)** | 50.5 (± 4.1) | **56.7 (± 4.7)** | **67.0 (± 7.0)** | 73.7 (± 11.4) |
| **Code Desc.** | | | | | |
| Observed | 22.0 (± 4.0) | **55.7 (± 3.3)** | **72.4 (± 3.2)** | **73.6 (± 3.4)** | **83.6 (± 1.7)** |
| Unobserved | **32.1 (± 7.2)** | 48.6 (± 7.2) | 63.3 (± 14.2) | 69.7 (± 12.4) | 72.9 (± 22.2) |
| **Hindu Know.** | | | | | |
| Observed | 25.1 (± 15.2) | **30.7 (± 2.6)** | 34.9 (± 0.9) | **52.4 (± 1.6)** | 53.9 (± 1.8) |
| Unobserved | **31.6 (± 10.7)** | 26.9 (± 4.4) | **37.5 (± 7.0)** | 47.1 (± 5.4) | **56.5 (± 3.5)** |
| **Known Unk.** | | | | | |
| Observed | 49.9 (± 1.9) | **66.9 (± 4.7)** | **76.2 (± 3.5)** | **79.3 (± 2.5)** | **84.7 (± 2.1)** |
| Unobserved | **52.8 (± 5.2)** | 63.8 (± 7.3) | 68.4 (± 11.1) | 69.0 (± 6.7) | 80.6 (± 8.1) |
| **Logical Ded.** | | | | | |
| Observed | 19.8 (± 0.7) | 27.1 (± 1.3) | 45.9 (± 1.1) | **52.5 (± 1.0)** | **58.0 (± 0.7)** |
| Unobserved | **19.9 (± 0.4)** | **28.9 (± 2.8)** | **46.4 (± 2.6)** | 52.1 (± 1.1) | 57.8 (± 0.6) |
| **Novel Conc.** | | | | | |
| Observed | **22.9 (± 9.2)** | 15.9 (± 5.4) | **31.0 (± 2.3)** | 29.8 (± 2.4) | **50.1 (± 1.9)** |
| Unobserved | 19.3 (± 3.6) | **16.8 (± 4.0)** | 28.4 (± 6.9) | **31.2 (± 5.0)** | 46.0 (± 5.4) |
| **Logic Grid** | | | | | |
| Observed | 22.3 (± 4.0) | **31.7 (± 0.8)** | 32.6 (± 2.1) | **41.8 (± 1.1)** | **43.2 (± 1.8)** |
| Unobserved | **28.8 (± 3.1)** | 29.4 (± 5.1) | **34.1 (± 2.8)** | 38.6 (± 5.4) | 39.6 (± 4.4) |
| **Conc. Com.** | | | | | |
| Observed | 30.4 (± 10.6) | **55.6 (± 5.1)** | **64.2 (± 1.9)** | **75.9 (± 1.8)** | **75.0 (± 2.6)** |
| Unobserved | **32.2 (± 16.7)** | 54.5 (± 9.4) | 58.1 (± 11.6) | 75.0 (± 1.9) | 73.6 (± 4.8) |

Table 22: Granular results corresponding to the subset reported in Figure 2b, on each dataset of category BBL-QA.

| BBL-BC | | | | | |
|---|---|---|---|---|---|
| **Model** | Flan-T5-XL | Flan-T5-XXL | T0pp-11B | Alpaca-7B | Alpaca-13B |
| **Play Dialog** | | | | | |
| OBSERVED | **61.6 (± 5.8)** | 51.8 (± 9.5) | **62.7 (± 0.4)** | **45.0 (± 2.0)** | **53.4 (± 5.8)** |
| UNOBSERVED | 53.0 (± 6.9) | **58.1 (± 4.4)** | 55.2 (± 8.1) | 42.9 (± 7.9) | 42.9 (± 8.8) |
| **Strat. QA** | | | | | |
| OBSERVED | 58.7 (± 3.3) | **64.2 (± 3.0)** | 51.0 (± 1.8) | 53.0 (± 2.1) | 56.7 (± 3.8) |
| UNOBSERVED | **60.7 (± 7.5)** | 59.3 (± 6.1) | **54.5 (± 0.9)** | **53.3 (± 4.1)** | **61.0 (± 1.9)** |
| **Strange St.** | | | | | |
| OBSERVED | 69.3 (± 4.4) | 71.0 (± 7.3) | **51.2 (± 5.1)** | **67.0 (± 4.7)** | **69.8 (± 5.0)** |
| UNOBSERVED | **70.5 (± 7.0)** | **77.4 (± 6.1)** | 48.4 (± 3.1) | 59.9 (± 9.4) | 57.5 (± 5.6) |
| **Winowhy** | | | | | |
| OBSERVED | **76.5 (± 1.9)** | **75.6 (± 4.0)** | **99.6 (± 1.0)** | 50.1 (± 4.8) | 51.9 (± 4.6) |
| UNOBSERVED | 60.2 (± 6.2) | 59.7 (± 7.7) | 60.9 (± 5.1) | **53.4 (± 4.6)** | **55.2 (± 6.0)** |

Table 23: Granular results corresponding to the subset reported in Table 3, on each dataset of category BBL-BC.

| BBL-BC | | | | | |
|---|---|---|---|---|---|
| **Size Variance** | Small (80M) | Base (250M) | Large (780M) | XL (3B) | XXL (11B) |
| **Play Dialog** | | | | | |
| OBSERVED | 51.6 (± 13.3) | 54.6 (± 10.7) | **59.0 (± 6.7)** | **61.6 (± 5.8)** | 51.8 (± 9.5) |
| UNOBSERVED | **61.6 (± 4.6)** | **56.3 (± 10.9)** | 57.3 (± 7.8) | 53.0 (± 6.9) | **58.1 (± 4.4)** |
| **Strat. QA** | | | | | |
| OBSERVED | **52.3 (± 1.0)** | 48.9 (± 2.1) | **60.9 (± 1.3)** | 58.7 (± 3.3) | **64.2 (± 3.0)** |
| UNOBSERVED | 51.5 (± 2.7) | **52.9 (± 1.3)** | 53.9 (± 3.8) | **60.7 (± 7.5)** | 59.3 (± 6.1) |
| **Strange St.** | | | | | |
| OBSERVED | 41.3 (± 10.3) | **43.1 (± 4.2)** | 54.4 (± 1.2) | 69.3 (± 4.4) | 71.0 (± 7.3) |
| UNOBSERVED | **55.9 (± 18.5)** | 42.0 (± 5.5) | **67.9 (± 8.0)** | **70.5 (± 7.0)** | **77.4 (± 6.1)** |
| **Winowhy** | | | | | |
| OBSERVED | **54.8 (± 1.6)** | 55.9 (± 7.6) | **60.4 (± 9.8)** | **76.5 (± 1.9)** | **75.6 (± 4.0)** |
| UNOBSERVED | 53.7 (± 5.1) | **57.1 (± 6.7)** | 53.5 (± 4.9) | 60.2 (± 6.2) | 59.7 (± 7.7) |

Table 24: Granular results for Figure 2b on each dataset of category BBL-BC

| BBL-MC | | | | | |
|---|---|---|---|---|---|
| **Model** | Flan-T5-XL | Flan-T5-XXL | T0pp-11B | Alpaca-7B | Alpaca-13B |
| **Language ID** | | | | | |
| OBSERVED | **32.6 (± 0.2)** | **38.9 (± 0.3)** | **15.7 (± 3.0)** | 12.9 (± 0.7) | 18.5 (± 0.7) |
| UNOBSERVED | 25.5 (± 7.3) | 31.6 (± 9.4) | 14.3 (± 2.4) | **14.7 (± 1.7)** | **21.7 (± 0.7)** |
| **Vitamin C** | | | | | |
| OBSERVED | 78.6 (± 1.1) | 78.5 (± 0.7) | 68.3 (± 1.1) | **51.4 (± 3.6)** | **54.9 (± 2.9)** |
| UNOBSERVED | 78.6 (± 3.6) | **80.2 (± 1.6)** | **68.5 (± 2.4)** | 18.1 (± 5.3) | 23.6 (± 14.2) |

Table 25: Granular results for Table 3 on each dataset of category BBL-MC

| BBL-MC | | | | |
|---|---|---|---|---|
| **Size Variance** | Small (80M) | Base (250M) | Large (780M) | XL (3B) | XXL (11B) |
| **Language ID** | | | | | |
| OBSERVED | **11.9 ($\pm$ 0.2)** | **17.0 ($\pm$ 0.3)** | **25.8 ($\pm$ 0.3)** | **32.6 ($\pm$ 0.2)** | **38.9 ($\pm$ 0.3)** |
| UNOBSERVED | 9.5 ($\pm$ 0.2) | 12.4 ($\pm$ 1.5) | 19.2 ($\pm$ 4.4) | 25.5 ($\pm$ 7.3) | 31.6 ($\pm$ 9.4) |
| **Vitamin C** | | | | | |
| OBSERVED | **46.6 ($\pm$ 4.0)** | 60.7 ($\pm$ 5.6) | **72.6 ($\pm$ 1.5)** | 78.6 ($\pm$ 1.1) | 78.5 ($\pm$ 0.7) |
| UNOBSERVED | 40.8 ($\pm$ 4.2) | **63.0 ($\pm$ 4.6)** | 36.4 ($\pm$ 0.8) | 78.6 ($\pm$ 3.6) | **80.2 ($\pm$ 1.6)** |

Table 26: Granular results for Figure 2b on each dataset of category BBL-MC

## K.2 ICL FEW-SHOT EXPERIMENT RESULTS

| MMLU (One-shot) | | | | | |
|---|---|---|---|---|---|
| **Flan-T5** | Small (80M) | Base (250M) | Large (780M) | XL (3B) | XXL (11B) |
| **MMLU** | | | | | |
| OBSERVED | 29.3 (± 0.9) | **33.9 (± 0.5)** | **40.7 (± 0.2)** | **47.5 (± 0.2)** | 52.7 (± 0.2) |
| UNOBSERVED | **29.6 (± 0.6)** | 33.8 (± 1.0) | 40.4 (± 0.9) | 47.5 (± 0.7) | **52.8 (± 1.0)** |

Table 27: Granular results on each dataset of MMLU for Flan-T5 models and one-shot ICL. We group all QA tasks in MMLU and report overall accuracy on these.

| BBL-QA (One-shot) | | | | | |
|---|---|---|---|---|---|
| **Flan-T5** | Small (80M) | Base (250M) | Large (780M) | XL (3B) | XXL (11B) |
| **BBQ Lite** | | | | | |
| OBSERVED | **29.6 (± 2.2)** | 50.5 (± 1.7) | 57.0 (± 2.0) | 66.7 (± 1.6) | 77.2 (± 2.7) |
| UNOBSERVED | 28.9 (± 1.4) | **51.0 (± 3.6)** | **58.0 (± 2.7)** | **69.0 (± 5.9)** | **77.6 (± 5.5)** |
| **Code Desc.** | | | | | |
| OBSERVED | 20.5 (± 3.1) | **56.9 (± 5.1)** | **76.0 (± 2.2)** | **73.6 (± 1.8)** | **85.3 (± 1.6)** |
| UNOBSERVED | **25.6 (± 8.7)** | 43.6 (± 8.4) | 53.5 (± 16.0) | 65.8 (± 15.3) | 75.0 (± 12.3) |
| **Hindu Know.** | | | | | |
| OBSERVED | **24.5 (± 12.7)** | **30.5 (± 3.6)** | 36.2 (± 1.7) | **50.9 (± 1.3)** | 52.9 (± 1.9) |
| UNOBSERVED | 23.0 (± 5.8) | 22.4 (± 5.8) | **37.7 (± 4.7)** | 50.2 (± 5.8) | **54.6 (± 3.2)** |
| **Known Unk.** | | | | | |
| OBSERVED | **49.2 (± 3.8)** | **66.7 (± 8.3)** | **73.6 (± 2.7)** | **76.3 (± 2.0)** | **84.7 (± 3.8)** |
| UNOBSERVED | 49.1 (± 4.1) | 60.2 (± 7.3) | 67.7 (± 12.9) | 63.7 (± 9.8) | 76.0 (± 12.7) |
| **Logical Ded.** | | | | | |
| OBSERVED | 20.2 (± 0.4) | 26.8 (± 1.0) | **45.9 (± 1.1)** | **53.0 (± 0.7)** | **58.2 (± 0.5)** |
| UNOBSERVED | **20.2 (± 0.8)** | **27.8 (± 3.5)** | 44.3 (± 8.6) | 48.6 (± 10.1) | 55.0 (± 10.5) |
| **Novel Conc.** | | | | | |
| OBSERVED | **21.3 (± 4.3)** | **14.8 (± 4.9)** | **29.1 (± 4.0)** | 31.2 (± 2.0) | **47.7 (± 3.1)** |
| UNOBSERVED | 17.2 (± 6.8) | 14.7 (± 9.0) | 24.7 (± 6.2) | **35.1 (± 5.2)** | 42.5 (± 9.0) |
| **Conc. Com.** | | | | | |
| OBSERVED | 28.0 (± 3.6) | **38.5 (± 2.3)** | **65.0 (± 1.8)** | **77.7 (± 1.6)** | **77.1 (± 2.2)** |
| UNOBSERVED | **28.6 (± 4.5)** | 36.3 (± 6.4) | 58.2 (± 10.7) | 74.8 (± 11.9) | 75.2 (± 2.8) |

Table 28: Granular results on each dataset of category BBL-QA with one-shot in-context learning.

| BBL-BC (One-shot) | | | | |
|---|---|---|---|---|
| **Flan-T5** | Small (80M) | Base (250M) | Large (780M) | XL (3B) | XXL (11B) |
| **Play Dialog** | | | | |
| OBSERVED | 49.8 (± 11.9) | **54.4 (± 10.3)** | **58.1 (± 5.7)** | **57.8 (± 3.3)** | 43.9 (± 4.2) |
| UNOBSERVED | **55.8 (± 10.2)** | 52.5 (± 10.7) | 48.6 (± 8.7) | 46.3 (± 3.9) | **51.3 (± 3.3)** |
| **Strat. QA** | | | | |
| OBSERVED | 52.5 (± 0.8) | 49.3 (± 3.1) | **60.6 (± 1.5)** | 60.6 (± 6.2) | **66.2 (± 2.6)** |
| UNOBSERVED | **53.2 (± 0.0)** | **53.3 (± 0.8)** | 55.9 (± 4.3) | **61.2 (± 6.0)** | 62.2 (± 5.5) |
| **Strange St.** | | | | |
| OBSERVED | 40.7 (± 12.3) | **41.8 (± 2.3)** | 51.7 (± 1.6) | 74.4 (± 3.6) | 78.5 (± 2.1) |
| UNOBSERVED | **46.7 (± 5.1)** | 37.9 (± 8.6) | **56.3 (± 3.0)** | **78.7 (± 3.2)** | **83.2 (± 7.6)** |
| **Winowhy** | | | | |
| OBSERVED | **52.7 (± 2.8)** | 57.3 (± 6.2) | **62.1 (± 5.9)** | **77.2 (± 0.6)** | **76.7 (± 1.0)** |
| UNOBSERVED | 48.1 (± 4.2) | **58.3 (± 8.1)** | 57.1 (± 8.8) | 66.3 (± 8.8) | 65.5 (± 9.9) |

Table 29: Granular results on each dataset of category BBL-BC with one-shot in-context learning.

| BBL-MC (One-shot) | | | | |
|---|---|---|---|---|
| **Flan-T5** | Small (80M) | Base (250M) | Large (780M) | XL (3B) | XXL (11B) |
| **Language ID** | | | | |
| OBSERVED | **11.7 (± 0.2)** | **13.5 (± 2.8)** | **25.6 (± 0.4)** | **31.8 (± 0.3)** | **38.7 (± 0.5)** |
| UNOBSERVED | 9.7 (± 0.9) | 11.6 (± 1.5) | 16.9 (± 5.4) | 20.3 (± 7.5) | 28.4 (± 10.3) |
| **Vitamin C** | | | | |
| OBSERVED | **50.9 (± 1.6)** | 60.9 (± 5.4) | **73.3 (± 0.8)** | **78.6 (± 1.4)** | 80.4 (± 0.5) |
| UNOBSERVED | 50.1 (± 0.9) | **64.5 (± 1.8)** | 73.2 (± 1.5) | 77.5 (± 9.3) | **80.7 (± 3.6)** |

Table 30: Granular results on each dataset of category BBL-MC with one-shot in-context learning.

## K.3 REPRESENTATION ALIGNMENT RESULTS

In the main paper we reported the aggregated results over benchmark corpora. We report means and stadard deviations of the accuracies achieved over all instructions in Table 31 and 32. We also share even more granular results—reporting the performance for each instruction—in CSV files provided in the supplemental material.

| | Flan-T5-XL | | | Alpaca-7B | | |
|---|---|---|---|---|---|---|
| | MMLU | | | | | |
| **Dataset** | OBS. | UNOBS. | Avg. | OBS. | UNOBS. | Avg. |
| **MMLU** | | | | | | |
| - Baseline | $48.0 \pm 0.3$ | $47.3 \pm 0.9$ | $47.7 \pm 0.6$ | $42.1 \pm 0.4$ | $40.6 \pm 1.6$ | $41.3 \pm 1.0$ |
| - Aligned | $\mathbf{48.4} \pm 0.2$ | $\mathbf{47.9} \pm 0.5$ | $\mathbf{48.2} \pm 0.6$ | $\mathbf{42.9} \pm 1.4$ | $\mathbf{41.7} \pm 2.0$ | $\mathbf{42.3} \pm 1.0$ |
| **Overall $\Delta$** | 0.4±0.0 | 0.6±0.0 | 0.5±0.0 | 0.8±0.0 | 1.1±0.0 | 1.0±0.0 |
| | BBL-MC | | | | | |
| **Dataset** | OBS. | UNOBS. | Avg. | OBS. | UNOBS. | Avg. |
| **Language ID** | | | | | | |
| - Baseline | $\mathbf{32.7} \pm 0.1$ | $29.2 \pm 1.0$ | $30.9 \pm 0.6$ | $12.6 \pm 0.9$ | $14.8 \pm 1.7$ | $13.7 \pm 1.3$ |
| - Aligned | $32.6 \pm 0.2$ | $\mathbf{29.4} \pm 0.8$ | $\mathbf{31.0} \pm 0.6$ | $\mathbf{12.9} \pm 0.5$ | $\mathbf{16.7} \pm 1.0$ | $\mathbf{14.8} \pm 1.3$ |
| **Vitamin C** | | | | | | |
| - Baseline | $77.6 \pm 0.0$ | $80.9 \pm 1.3$ | $79.3 \pm 0.7$ | $45.3 \pm 0.5$ | $13.3 \pm 0.1$ | $29.3 \pm 0.3$ |
| - Aligned | $\mathbf{78.3} \pm 0.1$ | $\mathbf{81.4} \pm 1.5$ | $\mathbf{79.8} \pm 0.7$ | $\mathbf{45.6} \pm 0.9$ | $\mathbf{18.5} \pm 0.4$ | $\mathbf{32.1} \pm 0.3$ |
| **Overall $\Delta$** | 0.3±0.4 | 0.3±0.1 | 0.3±0.2 | 0.3±0.0 | 3.5±1.6 | 1.9±0.8 |
| | BBL-BC | | | | | |
| **Dataset** | OBS. | UNOBS. | Avg. | OBS. | UNOBS. | Avg. |
| **Strat. QA** | | | | | | |
| - Baseline | $58.4 \pm 1.6$ | $59.9 \pm 8.0$ | $59.2 \pm 4.8$ | $53.4 \pm 2.0$ | $51.6 \pm 4.4$ | $52.5 \pm 3.2$ |
| - Aligned | $\mathbf{58.6} \pm 2.9$ | $\mathbf{60.6} \pm 6.6$ | $\mathbf{59.6} \pm 4.8$ | $\mathbf{54.6} \pm 3.1$ | $\mathbf{55.2} \pm 5.5$ | $\mathbf{54.9} \pm 3.2$ |
| **Strange St.** | | | | | | |
| - Baseline | $\mathbf{69.1} \pm 2.9$ | $69.4 \pm 4.1$ | $\mathbf{69.2} \pm 3.5$ | $69.6 \pm 1.3$ | $56.1 \pm 4.3$ | $62.9 \pm 2.8$ |
| - Aligned | $59.0 \pm 4.1$ | $\mathbf{69.8} \pm 5.0$ | $64.4 \pm 3.5$ | $\mathbf{70.5} \pm 1.1$ | $\mathbf{59.5} \pm 6.8$ | $\mathbf{65.0} \pm 2.8$ |
| **Play Dialog** | | | | | | |
| - Baseline | $58.0 \pm 10.9$ | $52.2 \pm 6.8$ | $55.1 \pm 8.9$ | $45.0 \pm 2.0$ | $45.6 \pm 10.8$ | $45.3 \pm 6.4$ |
| - Aligned | $\mathbf{61.4} \pm 3.0$ | $\mathbf{58.6} \pm 7.0$ | $\mathbf{60.0} \pm 8.9$ | $\mathbf{45.7} \pm 3.3$ | $\mathbf{48.8} \pm 11.8$ | $\mathbf{47.2} \pm 6.4$ |
| **Winowhy** | | | | | | |
| - Baseline | $76.2 \pm 1.9$ | $54.3 \pm 11.8$ | $65.3 \pm 6.9$ | $49.5 \pm 8.9$ | $55.7 \pm 0.1$ | $52.6 \pm 4.5$ |
| - Aligned | $\mathbf{77.6} \pm 3.3$ | $\mathbf{54.8} \pm 11.6$ | $\mathbf{66.2} \pm 6.9$ | $\mathbf{50.2} \pm 8.4$ | $\mathbf{58.6} \pm 1.9$ | $\mathbf{54.4} \pm 4.5$ |
| **Overall $\Delta$** | -1.3±5.2 | 2.0±2.5 | 0.4±3.5 | 0.9±0.2 | 3.2±0.3 | 2.1±0.2 |

Table 31: Granular results corresponding to the MMLU, BBL-MC, and BBL-BC subset reported in Table 7. Baseline refers to the performance of the original instruction-tuned model. We observe the significant improvement on following unobserved instructions consistently.

| | Flan-T5-XL | | | Alpaca-7B | | |
|---|---|---|---|---|---|---|
| | BBL-QA | | | | | |
| **Dataset** | OBS. | UNOBS. | Avg. | OBS. | UNOBS. | Avg. |
| **Code Desc.** | | | | | | |
| - Baseline | $75.0 \pm 3.1$ | $69.3 \pm 11.8$ | $72.2 \pm 7.5$ | $71.9 \pm 3.1$ | $65.6 \pm 14.7$ | $68.8 \pm 8.9$ |
| - Aligned | $\mathbf{76.4} \pm 1.8$ | $\mathbf{74.9} \pm 1.7$ | $\mathbf{75.7} \pm 7.5$ | $\mathbf{72.8} \pm 3.5$ | $\mathbf{71.7} \pm 17.3$ | $\mathbf{72.2} \pm 8.9$ |
| **Hindu Know.** | | | | | | |
| - Baseline | $52.7 \pm 2.4$ | $45.5 \pm 6.2$ | $49.1 \pm 4.3$ | $52.5 \pm 2.3$ | $49.0 \pm 5.3$ | $50.7 \pm 3.8$ |
| - Aligned | $\mathbf{54.0} \pm 2.6$ | $\mathbf{49.6} \pm 1.2$ | $\mathbf{51.8} \pm 4.3$ | $\mathbf{53.8} \pm 0.2$ | $\mathbf{55.9} \pm 1.5$ | $\mathbf{54.8} \pm 3.8$ |
| **BBQ Lite** | | | | | | |
| - Baseline | $\mathbf{66.7} \pm 1.9$ | $62.9 \pm 8.0$ | $64.8 \pm 5.0$ | $31.8 \pm 1.3$ | $33.2 \pm 0.6$ | $32.5 \pm 0.9$ |
| - Aligned | $66.0 \pm 2.3$ | $\mathbf{69.0} \pm 2.2$ | $\mathbf{67.5} \pm 5.0$ | $\mathbf{32.7} \pm 2.0$ | $\mathbf{36.3} \pm 2.2$ | $\mathbf{34.5} \pm 0.9$ |
| **Conc. Com.** | | | | | | |
| - Baseline | $75.7 \pm 1.5$ | $75.0 \pm 2.1$ | $75.3 \pm 1.8$ | $56.3 \pm 5.5$ | $53.1 \pm 6.2$ | $54.7 \pm 5.9$ |
| - Aligned | $\mathbf{81.4} \pm 1.1$ | $\mathbf{79.0} \pm 1.1$ | $\mathbf{80.2} \pm 1.8$ | $\mathbf{56.4} \pm 5.7$ | $\mathbf{56.1} \pm 6.9$ | $\mathbf{56.3} \pm 5.9$ |
| **Logical Ded.** | | | | | | |
| - Baseline | $52.6 \pm 0.3$ | $52.4 \pm 0.7$ | $52.5 \pm 0.5$ | $25.7 \pm 0.8$ | $25.4 \pm 1.4$ | $25.5 \pm 1.1$ |
| - Aligned | $\mathbf{52.6} \pm 0.7$ | $\mathbf{52.8} \pm 0.6$ | $\mathbf{52.7} \pm 0.5$ | $\mathbf{25.9} \pm 0.5$ | $\mathbf{28.4} \pm 1.2$ | $\mathbf{27.2} \pm 1.1$ |
| **Novel Conc.** | | | | | | |
| - Baseline | $\mathbf{31.9} \pm 2.6$ | $\mathbf{32.5} \pm 4.2$ | $\mathbf{32.2} \pm 3.4$ | $36.9 \pm 4.1$ | $35.0 \pm 2.6$ | $35.9 \pm 3.3$ |
| - Aligned | $28.7 \pm 2.0$ | $31.6 \pm 4.3$ | $30.2 \pm 3.4$ | $\mathbf{37.5} \pm 3.9$ | $\mathbf{39.2} \pm 2.4$ | $\mathbf{38.3} \pm 3.3$ |
| **Logic Grid** | | | | | | |
| - Baseline | $42.0 \pm 0.9$ | $40.0 \pm 4.4$ | $41.0 \pm 2.7$ | $25.7 \pm 0.9$ | $25.6 \pm 1.5$ | $25.7 \pm 1.2$ |
| - Aligned | $\mathbf{45.1} \pm 3.6$ | $\mathbf{41.1} \pm 4.1$ | $\mathbf{43.1} \pm 2.7$ | $\mathbf{26.1} \pm 1.0$ | $\mathbf{30.4} \pm 4.2$ | $\mathbf{28.2} \pm 1.2$ |
| **Known Unk.** | | | | | | |
| - Baseline | $78.3 \pm 1.5$ | $\mathbf{67.4} \pm 6.3$ | $\mathbf{72.8} \pm 3.9$ | $72.9 \pm 4.7$ | $61.3 \pm 9.3$ | $67.1 \pm 7.0$ |
| - Aligned | $\mathbf{79.3} \pm 1.6$ | $65.5 \pm 10.0$ | $72.4 \pm 3.9$ | $\mathbf{74.0} \pm 4.3$ | $\mathbf{67.4} \pm 7.1$ | $\mathbf{70.7} \pm 7.0$ |
| **Overall** $\Delta$ | $1.1 \pm 2.5$ | $2.3 \pm 2.8$ | $1.7 \pm 2.1$ | $0.7 \pm 0.4$ | $4.6 \pm 1.5$ | $2.7 \pm 0.9$ |

Table 32: Granular results corresponding to the BBL-QA subset reported in Table 7. Baseline refers to the performance of the original instruction-tuned model. We observe the significant improvement on following unobserved instructions consistently.

