# OpenReview forum: "Evaluating the Zero-shot Robustness of Instruction-tuned Language Models"
_ICLR.cc/2024/Conference — ICLR 2024 spotlight_

### Official Review · Reviewer_KbYU · 2023-10-24

**Soundness:** 3 good
**Presentation:** 3 good
**Contribution:** 4 excellent
**Rating:** 8
**Confidence:** 4

**Summary:**

The study examines the robustness of LLMs in zero-shot prompting. It was observed that the model's performance varies based on the instructions used, particularly with unfamiliar instructions. To mitigate this decline in performance, the study proposes a method that incorporates soft-prompts and trains the model with an additional loss function designed to align the embedding representation of the instructions.

**Strengths:**

This paper provides a thorough comparison of LLM's instruction robustness across various models and sizes. Although the finding—that LLMs are not robust to varying instructions—is anticipated, this paper presents compelling empirical evidence. Additional analysis in the embedding space of the instruction is interesting. Furthermore, the study suggests an elegant solution to narrow the performance disparity between familiar and unfamiliar instructions.

**Weaknesses:**

The proposed solution could benefit from more details. Exploring the soft-prompt size (i.e., the variable N) would be nice. The N used is not specified in the main text either, which I believe is an important detail.

They provide synthetic instruction, and one of the baselines is to fine-tune normally with this new instruction. Interestingly, there is a noticable degradation in performance, yet there's no explanation as to why. Usually, continual fine-tuning on the same data should not damage performance (significantly so). My guess is that the quality of the synthetic prompt is not very good. If so, this raises a question: what if we spend more effort on generating better synthetic instruction beforehand?

**Questions:**

- "We then form batches by including one instance featuring the original instruction, and the rest comprising paraphrased instructions." -> what's the batch size, and consequently, what's the paraphrase size and the final dataset size altogether?
- Significant degradation on FT on Table 7, why is that so?

---

> ### Author Response · Authors · 2023-11-20
> **Response to KbYU**
>
> **W1 (Soft-prompt size)**: We use a prefix length of 10 for the result reported (thank you for pointing this out!) We have explored many different settings (e.g., soft-prompt size) in our pilot studies and noticed no substantial differences, suggesting that one needs only a minimal set of free parameters to tune with the proposed objective to realize improvements. Below we reproduce results observed when using different soft-prompt sizes (from 1 to 50), and show their performance and semantic distance:
>
> | **Soft Prompt Size** | **MMLU Obs.**  | **MMLU Unobs.** | **MMLU Ave.**  | **Ave. MMLU Distance** | **BBL Obs.**    | **BBL Unobs.**  | **BBL Ave.**    | **Ave. BBL Distance** |
> |----------------------|----------------|-----------------|----------------|------------------------|-----------------|-----------------|-----------------|-----------------------|
> | 1                    | **47.4 ± 0.2** | **47.8 ± 0.2**  | **47.6 ± 0.3** | 3.82                   | 55.9 ± 19.7     | 54.2 ± 17.6     | 55.1 ± 18.4     | 7.33                  |
> | 5                    | 47.9 ± 0.2     | 48.4 ± 0.3      | 48.1 ± 0.3     | 4.03                   | 55.8 ± 21.2     | **52.4 ± 19.0** | **54.1 ± 19.9** | **6.88**              |
> | 10                   | 47.6 ± 0.1     | 48.0 ± 0.2      | 47.8 ± 0.3     | 4.30                   | **55.0 ± 21.6** | 53.6 ± 18.0     | 54.3 ± 19.6     | 7.47                  |
> | 30                   | 47.6 ± 0.1     | 48.2 ± 0.1      | 47.9 ± 0.3     | 3.84                   | 55.4 ± 20.8     | 53.8 ± 17.9     | 54.6 ± 19.3     | 7.02                  |
> | 50                   | 48.0 ± 0.2     | 48.6 ± 0.2      | 48.3 ± 0.5     | **3.73**               | 55.9 ± 21.9     | 52.6 ± 18.9     | 54.2 ± 20.0     | 7.03                  |
>
> The results indicate that the prefix length does not affect the performance for both observed and unobserved instructions. As they fluctuate around the same level. Additionally, we also calculated average distance between unobserved instructions to their closest equivalent observed instructions, and the value is highly similar between all soft prompt sizes.
>
> The takeaway here is that soft-prompt is only a method we use to drag the equivalent instructions close together without affecting the model’s performance, and the number of parameters we dedicate to do that is seemingly insignificant.
>
> **Q1 (Dataset size)**: The batch size we use for both models is 4, hence each batch contains one “prototype” instruction and 3 paraphrased instructions to align (this is also reported at the beginning of Appendix B.3.) We randomly sampled 1k prototype samples from the original instruction tuning datasets, hence the final dataset altogether is 4k; we report this Section 5.
>
> **W2 & Q2 (Significance Test)**: Please see our general response, (1). Briefly, this is likely an instance of “catastrophic forgetting”, a hypothesis supported by the results we have added in Appendix G.

---

### Official Review · Reviewer_hcVs · 2023-10-28

**Soundness:** 2 fair
**Presentation:** 3 good
**Contribution:** 3 good
**Rating:** 6
**Confidence:** 4

**Summary:**

This paper observes that current instruction-tuned LLMs are sensitive to surface forms of instructions by showing that they underperform on unseen instructions compared to seen instructions. To mitigate this issue, the paper proposes soft prompt alignment technique.

**Strengths:**

- The paper performs experiments on various tasks including reasoning benchmarks (MMLU and BBL) and NLG benchmarks.
- The paper provides the experimental setting of evaluating the robustness of LLMs on different instructions by releasing various instructions collected from NLP researchers, which might facilitate further research.

**Weaknesses:**

- In Tables 3 and 7, compared to the performance difference, the variance is very large, questioning the significance of the difference between observed and unobserved instructions. A significance test should be conducted to test the significance of the difference.
- Although the title, abstract, and motivation of the paper mention that the paper tries to evaluate the zero-shot robustness of instruction-tuned language models generally, it only focuses on models up to 11B. The tendency is likely to be unpredictable for state-of-the-art LLMs such as GPT-3.5 or GPT-4.
- How does PT+KL on Table 7 perform on negated instructions? The alignment stage might make the LLM insensitive to most instructions even though the instructions are not relevant or negated, which is a similar finding for instruction-tuned LLMs for Webson & Pavlick (2022)

Webson & Pavlick (2022) - Do Prompt-Based Models Really Understand the Meaning of their Prompts?

**Questions:**

- Do you think that RLHF after instruction tuning might mitigate this issue?

---

> ### Author Response · Authors · 2023-11-20
> **Response to hcVs (1/2)**
>
> We appreciate the thoughtful feedback.
>
> **W1 (Significance test)**: Please see our general response, (2). Briefly: The results in Tables 3 and 7 are statistically significant, despite the observed variances. We have added significance results to the manuscript. We also note that in Table 3, the variances are much larger for the unobserved instructions, which is part of the issue that we are highlighting.
>
> **W2 (Size of LLMs)**: The reviewer is correct that our work here is focussed on modestly sized LMs. We have tried to be clear about this in the abstract and introduction. We write in the second sentence in the abstract that instruction tuning “... has shown particular strength in improving the performance of modestly sized LLMs, sometimes inducing performance competitive with much larger model variants”; this is what motivates our focus on these <= 11b models. We believe this analysis is important precisely because prior work has suggested that instruction tuning imbues such models with performance surprisingly competitive with massive LLMs; our findings call this into question.
>
> **W3 (Negation)**: The reviewer suggests an interesting additional analysis: Evaluating sensitivity to instruction negation. We have now conducted such analyses and report details (experimental setup and results) in Appendix F. For convenience, below we reproduce the aggregated results on how PT + KL does on all 6 settings (including negated observed instructions) we conducted in Section 4.5
>
> | **Method**| Closest | Incorrect | Negated | Task Designer | Collected | Nonsensical |
> |-|-|-|-|-|-|-|
> | - Baseline | 58.1 $\pm$ 1.8     | 52.2 $\pm$ 8.6 | 48.3 $\pm$ 3.8 | 44.2 | 55.3 $\pm$ 4.5 | 30.6 $\pm$ 4.4 |
> | - PT + KL | **60.5 $\pm$ 2.4** | **54.2 $\pm$ 7.2** | 47.1 $\pm$ 4.9 | **47.1** | **56.7 $\pm$ 3.9** | 38.0 $\pm$ 4.1 |
> | - PT + KL (Negation) | 58.6 $\pm$ 2.0     | 52.7 $\pm$ 9.2 | **43.0 $\pm$ 4.9** | 46.5 | 55.6 $\pm$ 4.3 | 35.7 $\pm$ 5.6 |
>
> This table reports the results we reported in Figure 3 (first row) after applying PT + KL (second row). We observe that PT + KL yields non-trivial improvement on models performance with: 1) suitable instructions (Closest) that are observed. 2) suitable instructions (Task Designer, Collected) that are unobserved. However, results also indicate that the proposed method does not succeed fully in terms of imbuing the model with the ability to truly “understand” instructions:
> The performance of observed instructions after adding negation does not drop significantly (only by 1.2 on average)
>
> We hypothesize that training the model to align semantically equivalent instructions does not explicitly incent the recognition of negated prompts. This is because our objective enforces that equivalent instructions should be similar, but negations rarely occur in the collected paraphrases. Moreover, learning to handle negations would more intuitively be done by encouraging the model to push negated versions of instructions away from their original forms; this differs from the form of our objective which focuses on similarity. Therefore, we conduct another experiment by modifying the objective slightly to handle negation.
>
> Specifically, we trained the model to push the representation of the negated instructions away from the original ones. To do so, we take the same 1k seed instances from the FLAN collection and prompt (with three different prompts) GPT-4 to rewrite them such that the instruction is negated. We keep all the hyperparameters same as how we did for PT + KL, but use KL to push the representation of negated instructions away from the originals:
>
> $\mathcal{L} = (1-\lambda) \mathcal{L}_{CE} - \lambda \mathcal{L}$$ _{KL}$
>
> After training, we run the “closer look” experiment again to see if this slightly different objective would result in a stronger robustness (third row). The result shows a significant drop of the accuracy when negation is added to the observed instruction. However, the performance in other settings remains comparable with the baseline when applying the PT + KL (Negation).
>
> This result provides additional evidence for our findings that robustness to instruction variants is strongly tied to the distances between learned representations of instructions: When we incentivize the model to learn similar representations for equivalent instructions, it performs better when given equivalent unobserved instructions, though it does not better handle negated instructions. But if we modify this slightly to encourage the model to induce dissimilar representations between instructions and their negated versions, it improves its responsiveness to negated instructions. In short, the proposed strategy of targeting (dis)similarity of model instruction representations is general enough to facilitate realizing robustness beyond paraphrases.
>
> We report a full set of results for this finding in Appendix F in the updated draft.

---

> ### Author Response · Authors · 2023-11-20
> **Reponse to hcVs (2/2)**
>
> **Q1 (RLHF)**: RLHF is most appropriate for aligning model outputs for tasks that require open-ended responses. In our work, we are benchmarking instructions and tasks that have a set of correct answers. Collecting human feedback for training is not a good fit for our main evaluation measures, which focus exclusively on the correctness of the output for (mostly) “objective” tasks like classification.
>
> However, the reviewer raises an interesting idea, in that it is possible that optimizing LLMs to align with human preferences generally via RLHF may implicitly require models to “understand” instructions in a more robust way.
>
> We perform a preliminary investigation into whether training with RLHF on an unrelated, open-ended text generation tasks affects the robustness of the instruction-following abilities. Specifically, we take an off-the-shelf Alpaca-7B trained with RLHF via PPO and run it on all the tasks and instructions we have used in this paper. We found that the output tends to do “extra things” for a given instruction (e.g., providing its explanation for question answer), we found that the overall performance is highly unstable and degrades significantly in most of the datasets.
>
> | **Categories** | Delta Obs. Accuracy | Delta Unobs. Accuracy | Delta Ave. Accuracy |
> |-|-|-|-|
> | **MMLU**| -18.5 ± 0.5 | -17.4 ± 1.7 | -18.0 ± 1.1|
> | **BBL-QA**| -11.2 ± 4.0 | -9.7 ± 4.7 | -10.4 ± 4.3|
> | **BBL-MC**| -0.3 ± 0.8 | -5.3 ± 0.8|  -2.8 ± 0.8 |
> | **BBL-BC**| -11.8 ± 1.7 | -6.2 ± 8.3| -9.0 ± 5.0 |
>
> Therefore, we do not have evidence to show that RLHF on modestly-sized LLMs would mitigate such issue.

---

> > ### Comment · Reviewer_hcVs · 2023-11-22
> > **Further question on author response**
> >
> > Hi, thank you for the response and interesting new results.
> >
> > I have one further question regarding my concerns.
> > Regarding W3, do you expect joint training on both objectives (PT+KL & PT+KL (Negative)) could enhance the performance for all cases?
> >
> > Typo Error: Table 3 caption -> I think the caption should be fixed into p < 0.05 instead of p < 0.5.

---

> > > ### Author Response · Authors · 2023-11-22
> > > **RE: Further question on author response**
> > >
> > > Hi, thanks for engaging, and for the additional feedback!
> > >
> > > We fixed the typo in Table 3 (thank you for pointing this out).
> > >
> > > The suggestion to combine the objectives aimed at general robustness via paraphrase alignment (proposed in the paper) and the negation robustness term (suggested by the reviewer and implemented in this response period) is clever! We have run a preliminary experiment with this “joint loss”. Details follow, but briefly: The preliminary results are promising, in that we see some gains in terms of robustness to rephrasing and negations, respectively, but these are reduced compared to the variants in which we optimize for only one of these (i.e., including only the paraphrase objective results in better performance on paraphrased instructions than when this is mixed with the negation objective). That said, given the timeframe, our investigation was necessarily limited here; mixing these objectives is an interesting direction for future work.
> > >
> > > **Implementation details:** Due to the fact that training to align the instructions (PT + KL) and training to push away negations (PT + KL (Negative)) use different augmented data, we cannot combine the losses over instances. Therefore, we instead mixed the datasets together, and assigned a “similarity” score $S$ to each instruction instance: 1 for equivalent instructions, and -1 for opposite (negated) instructions, with their respective prototypes. Then for each batch with the prototype instructions and mixture of its paraphrased/negated instructions, we optimize the modified KL loss:
> > >
> > > $\mathcal{L}_{KL} =\frac{1}{N-1} \sum_i^N \sum_j S_j KL(\hat{y}_i^j || \hat{y}_r^j)$,  $i\neq r$
> > >
> > >
> > > Where $S_j$ is the similarity score of the instructions with value equals either -1 (negation) and 1 (alignment). We then use the exact same procedures and hyperparameters to optimize the loss, and the total training data is essentially:
> > >
> > > $|D|=|D_{align}| + |D_{negate}| $
> > >
> > > The new result is reported in the table (This will also be added to the next revision of the appendix)
> > >
> > > | **Method**| Closest | Incorrect | Negated | Task Designer | Collected | Nonsensical |
> > > |-|-|-|-|-|-|-|
> > > | - Baseline | 58.1 $\pm$ 1.8     | 52.2 $\pm$ 8.6 | 48.3 $\pm$ 3.8 | 44.2 | 55.3 $\pm$ 4.5 | 30.6 $\pm$ 4.4 |
> > > | - PT + KL | **60.5 $\pm$ 2.4** | **54.2 $\pm$ 7.2** | 47.1 $\pm$ 4.9 | **47.1** | **56.7 $\pm$ 3.9** | 38.0 $\pm$ 4.1 |
> > > | - PT + KL (Negation) | 58.6 $\pm$ 2.0     | 52.7 $\pm$ 9.2 | **43.0 $\pm$ 4.9** | 46.5 | 55.6 $\pm$ 4.3 | 35.7 $\pm$ 5.6 |
> > > | - PT + KL (Joint) | 59.3 $\pm1.7$ | 52.8 $\pm8.7$ | 44.2 $\pm4.5$ | 46.6  | 56.1 $\pm3.5$ | 36.1 $\pm5.1$ |
> > >
> > > **Result**: Training with the joint loss does realize some of the advantages of both negative and paraphrase alignment methods. Compared with the standard fine-tuned baseline (no KL), we see consistent improvements across all settings. But we see stronger improvements in the respective settings when we optimize for only paraphrases or negations alone.
> > >
> > > **Conclusion:** Combining these losses comprehensively improves model robustness, as the reviewer hypothesized. In addition, this result indicates a relatively neutral affinity between the models’ ability to follow semantically equivalent instructions and ability to recognize negation, suggesting that the underlying mechanism of the robustness to these two aspects may be different.
> > >
> > > We think this highlights an interesting research direction toward improving this comprehensive robustness, perhaps via better data filtering, simply hyperparameter tuning, and other strategies to more carefully combine these losses.

---

> ### Author Response · Authors · 2023-11-23
> **Extended Analysis on Scaling for Reviewer hcVs**
>
> As a final additional response to reviewer feedback, we extend our analysis a bit to include a larger (20B) model. As reviewer hcVs pointed out, readers may be interested to see if the results generalize to (very) large models. While we maintain our stance that the success of instruction-tuning for boosting the performance of modestly-sized LLMs is the most exciting aspect of this method, and therefore worthy on its own of study, we nonetheless agree that studying the relationship between robustness and scale of such models is interesting.
>
> However, we can draw valid comparisons only if we keep all other variables (model architecture, pretraining data, and instruction tuning set) fixed. Therefore, we are not able to study black-boxed models (like GPT-3.5 or GPT-4) using this approach. To include additional scale in our work, we therefore conduct the full experiments on different sizes of Flan-T5 ranging from 70M to 11B (Fig 2a and Sec 4.3), as well as Flan-UL2 (20B), and we report the change of **delta performance** (observed - unobserved) in the table below:
>
> | **Model**     | # Parameters | MMLU               | BBL-QA            | BBL-MC             | BBL-BC              |
> |---------------|--------------|--------------------|-------------------|--------------------|---------------------|
> | Flan-T5-Small | 70M          | **-2.6 $\pm$ 8.3** | 4.1 $\pm$ 3.1     | **-0.4 $\pm$ 1.4** | **-7.9 $\pm$ 12.0** |
> | Flan-T5-Base  | 250M         | 0.0 $\pm$ 0.9      | **1.6 $\pm$ 6.2** | 1.2 $\pm$ 4.3      | 1.5 $\pm$ 8.3       |
> | Flan-T5-Large | 770M         | 0.1 $\pm$ 0.5      | 3.1 $\pm$ 6.4     | 11.4 $\pm$ 2.1     | 1.6 $\pm$ 6.1       |
> | Flan-T5-XL    | 3B           | 0.6 $\pm$ 0.7      | 3.0 $\pm$ 4.6     | 3.5 $\pm$ 3.2      | 5.5 $\pm$ 7.8       |
> | Flan-T5-XXL   | 11B          | 0.5 $\pm$ 0.6      | 3.4 $\pm$ 5.7     | 2.8 $\pm$ 3.1      | 2.0 $\pm$ 9.1       |
> | Flan-UL2      | 20B          | 0.2 $\pm$ 0.5      | 3.4 $\pm$ 7.3     | 0.6 $\pm$ 0.3      | 1.9 $\pm$ 7.1       |
>
> The results above (aggregated by task category) show that there is no apparent relationship between model size and the delta performance (our measure of robustness).
>
> More scientifically, we perform linear regression on the size of the model v.s. (observed  - unobserved performance) over data points from all datasets, this yields a slope of $4.45 \times 10^{-13}$ with a p-value of 0.087 ($R^2$ is 0.002). This suggests that model size may weakly correlate with increased robustness, but this relationship is not significant the $p=0.05$ level, and the relationship is tenuous. In other words, scale does seem to improve robustness, but only very moderately.
>
> We hope the additional results (together with those posted earlier) answer your questions. We ask that you consider whether it might be appropriate to adjust your score, if you feel these new findings have changed your view on the work. Thank you.

---

> > ### Comment · Reviewer_hcVs · 2023-11-23
> > **Response to authors**
> >
> > Thank you for your update.
> > I have increased my score accordingly.
> > I would appreciate it if the authors could update the paper with all the new results during the discussion period.

---

> > > ### Author Response · Authors · 2023-11-23
> > > **RE: Response to authors**
> > >
> > > Hi, thanks for engaging discussion!
> > >
> > > We have updated the paper with all the new results in the discussion period. Specifically, in addition to the changes mentioned in "**Summary of Changes to Paper**", we have made the following new updates:
> > >
> > > Main paper:
> > > - Table 3 typo fixed: p < 0.05 instead of p < 0.5
> > > - New mention of the extra results of scaling (Appendix H) in Sec. 4.3
> > >
> > > Appendix:
> > > - **Appendix F.1 on page 22**: add the results and discussion of the joint loss
> > > - New Section **Appendix H on page 24**: results and discussion of the scaling experiments including Flan-UL2
> > >
> > > Thank you.

---

### Official Review · Reviewer_cGUu · 2023-10-29

**Soundness:** 3 good
**Presentation:** 3 good
**Contribution:** 3 good
**Rating:** 8
**Confidence:** 4

**Summary:**

The authors analyse how sensitive various instruction-tuned models are to rephrased instructions, especially in zero-shot settings. They find that using novel instructions previously unseen by the model results in degraded performance across different tasks, and does not seem to improve with model scale. Further analysis suggests that instructions that are more dissimilar to instructions seen during training generally perform (somewhat) poorer. Finally, they introduce a prompt-tuning method that utilises a KL penalty to punish representations that drift from paraphrased versions of instructions. They show this performs better than simply training on the paraphrased instructions alone.

**Strengths:**

- The experiments are clear and well-thought-out, and the results are solid and interesting, affecting many current popular models (Llama finetunes), and holding for different model families.
- The observed correlations between performance and instruction similarity (in representation space) are interesting (even if the correlation is somewhat weak).
- The proposed fix method is well-motivated, and the results seem reasonable, especially for the Alpaca-7B model.
- The robustness of models to different prompts is an important and interesting area of study, especially considering current difficulties with LM evaluations.

**Weaknesses:**

- The novelty of the robustness experiments is somewhat limited, as there is prior work that has observed variance based on prompt formatting (discussed in the work itself). Considering this work seems to examine more modern models such as Alpaca, and considers a different set of tasks, I think this is a minor weakness - the work does add a number of interesting insights, especially around representation distance.
- The claim that models perform worse at novel phrasings of tasks seems a bit strong compared to the results: examining Table 3, the variance of performance on unobserved phrasings often is greater than the difference in average performance, suggesting that the difference in performance may not be significant (although the large increase in variance is certainly not desirable and interesting to see).
- While the proposed method seems strong, examining the results in Appendix B.3 seems to show that in many cases, prompt-tuning alone performs within one standard deviation of the PT+KL results (e.g., for BBL unobserved and average). As such, the proposed method might not be significantly different from just prompt tuning.

Overall, I think this paper is okay, and the robustness results are interesting and useful to see, along with the representation alignment results. However, the novelty of the robustness experiments is a bit limited, and it’s not clear the proposed method for improving this is significantly better than baselines.

**Questions:**

- How does your proposed method compare to introducing the model paraphrases during initial instruction finetuning? Training on the paraphrased versions alone after the instruction tuning may cause forgetting in an undesirable way.
- Is the degradation of performance on unobserved prompts statistically significant? The variance seems very high.
- Doing prompt tuning alone seems to often perform very similarly to PT+KL, and within the reported standard error in Appendix B.3. Do you have statistical tests for significance for your results?
- It would be interesting to see the results in Table 7 broken down into the BBL QA/BC/MC categories shown in Table 6 to see how much the difference in closest distance affects the performance of these different categories.

---

> ### Author Response · Authors · 2023-11-20
> **Response to cGUu**
>
> We thank the reviewer for their thoughtful comments.
>
> **W1 (Novelty)**: Regarding novelty, please see our general response, (3). In brief, our work builds upon but complements and considerably deepens these prior efforts.
>
> **Q1 (Augment to initial instruction-tuning set)**: Due to the modest size of the paraphrases set (4k), it has minimal effect on the training performance and is very time-consuming by adding to the initial instruction finetuning. Please refer to Table 11 in Appendix B.3.1 for the detailed comparison. We reiterate that the main goal of the improvement section is to find a way to narrow down the semantic gap between the observed and unobserved, so that we validate our hypothesis (representation similarity for equivalent instructions v.s. robustness )
>
> **W2 & Q2 (Significance Test)**: Please see our general response, (2). Briefly, the large variances here in part reflect the fact that we are reporting aggregated results over large benchmarks. That said, one of the main observations is that variances are considerably larger for unobserved instructions, which is—as the reviewer rightly points out—undesirable.
>
> To the reviewers’ question: The results in aggregate are statistically significant (p << 0.05) under a paired t-test, and if we run the t-test separately for each model (which avoids the issue of repeating datasets in observations), we observe p < 0.05 for all but T0++ (p=0.13).
>
> **W3 & Q3 (Efficacy of the objective)**: We observe two major differences between the performance using just prefix tuning (PT) and prefix tuning with extra objectives (PT + KL).
> The performance diff between unobserved instructions and observed instructions - i.e., robustness - is significantly higher (close to zero) for PT + KL (Table 7 in Section 6)
> The average standard deviation of the instruction performance of PT is even higher than the baseline, indicating instability in its performance; whereas this high variance is significantly mitigated with PT + KL (**Table 10 in Appendix B.3**)
>
> Intuitively, we design the extra objective to explicitly “increase the similarity” between equivalent instructions — unobserved to observed — from the model’s “perspective”, which we hypothesize to have positive correlation with the robustness of instruction following ability.
>
> Motivated by this, we calculate the average l2 distance between unobserved instructions and its closest observed instruction over all the examples (Same approach as Table 6) for PT and PT + KL.  (Fig 7 in Appendix B.3.2). We show that in 13 out of 14 datasets, the distance between equivalent instructions is smaller for PT + KL. The consistency of the close distance and better robustness supports our hypothesis of the correlation.
>
> We have added these results, figures and the discussion to Appendix B.3.2 of the revised paper.
>
> **Q4 (Disaggregated results for Table 7)**: We appreciate the suggestion and we have now added the disaggregated result in a dataset level in Table 30 and Table 31 in Appendix J.3. For convenience, here we reproduce results disaggregated by categories:
>
> | Categories | Alpaca Obs. Delta | Alpaca Unobs. Delta | Alpaca Ave. Delta | Flan Obs. Delta | Flan Unobs. Delta | Flan Ave. Delta |
> |------------|-------------------|---------------------|-------------------|-----------------|-------------------|-----------------|
> | MMLU       | 0.4               | 0.6                 | 0.5               | 0.8             | 1.1               | 1.0             |
> | BBL-QA     | 1.1 $\pm$ 2.5     | 2.3 $\pm$ 2.8       | 1.7 $\pm$ 2.1     | 0.7 $\pm$ 0.4   | 4.6 $\pm$ 1.5     | 2.7 $\pm$ 0.9   |
> | BBL-MC     | 0.3 $\pm$ 0.4     | 0.3 $\pm$ 0.1       | 0.3 $\pm$ 0.2     | 0.3 $\pm$ 0.0   | 3.5 $\pm$ 1.6     | 1.9 $\pm$ 0.8   |
> | BBL-BC     | -1.3 $\pm$ 5.2    | 2.0 $\pm$ 2.5       | 0.4 $\pm$ 3.5     | 0.9 $\pm$ 0.2   | 3.2 $\pm$ 0.3     | 2.1 $\pm$ 0.2   |
>
> This Table reports the average delta performances observed after applying the proposed method on Flan-T5-XL and Alpaca-7B. For BBL datasets we report standard deviations over the individual datasets (for MMLU we group all together so do not report a variance).
> Table 6 in the paper reports the average distance change after applying the objectives for Flan-T5-XL. In general, we observe a consistent but weak correlation relating the closest distance between equivalent instructions to the delta performance for unobserved instructions. The similarity between equivalent instructions changes little for MMLU, which leads to a marginal improvement. However, this pattern is also not absolute; the BBL-QA category in general receives the greatest improvement with respect to performance but not distance. We leave further analyses of instruction representations and downstream performance to future work building on our results here.

---

> > ### Comment · Reviewer_cGUu · 2023-11-21
> >
> > Hi, thanks for your response!
> >
> > Augmenting the instruction-set - When you say "the model is retrained from scratch" in Table 11, what is the setting here? Did you retrain from Llama-7B/T5-XL with the paraphrases and instructions mixed, or just train on the paraphrase data?
> > I agree that investigating representation similarity for equivalent instructions v.s. robustness is interesting, but I was primarily curious if using paraphrased data in a simple way during instruction-tuning itself increases robustness and increases the representation similarity in a simple way.
> >
> > Thank you for the significance tests and the new results. I agree that the reduction in variance using PT+KL is interesting and useful! I've read the other reviews and responses, and am bumping up my score, since the significance tests cover my main concern around the efficacy of the proposed method.

---

> > > ### Author Response · Authors · 2023-11-21
> > > **RE: Official Comment**
> > >
> > > Hi, thanks for engaging, and for the additional feedback!
> > >
> > >
> > > In Table 11, we initialized to the checkpoint weights "google/t5-v1.1-xl" and "llama-7", which are the foundation models for Flan-T5-XL and Alpaca-7B, respectively. We fine-tuned these models with the mixture of the original instruction training set and acquired paraphrases, following the exact procedure as used for Alpaca and everything but the packing technique (putting multiple instances into one forward pass) for Flan-T5-XL. This, to us, seemed like the simplest way to use the paraphrased data in instruction-tuning.
> > >
> > >
> > > The results in Table 11 indicate that this does not provide much benefit, relative to the original instruction-tuned model (for Flan-T5; we see a slight improvement for Alpaca). We hypothesize that including the additional instructions in a standard instruction-tuning setup probably yields little benefit because the added set of new instructions/paraphrases is relatively small (e.g., 1/52 of the Alpaca dataset size); adding the KL term to explicitly pair these provides additional signal, and—as we show—induces similar representations for equivalent instructions in a way that naively training on all data does not. If one collected a much larger set of paraphrases, fine-tuning in the straightforward way may prove to yield more substantial benefits.
> > > We hope that this answers your question, but please let us know if we can clarify further. Thank you again for your revised feedback!

---

> > > > ### Comment · Reviewer_cGUu · 2023-11-21
> > > >
> > > > Thanks for the clarification, that finetuning setup makes sense! It would definitely be interesting to scale up the paraphrase set, but I think that's a good thing to leave for future work. Showing that you can make use of a small amount of data more effectively using your method is a nice contribution.

---

### Official Review · Reviewer_RNd8 · 2023-11-01

**Soundness:** 3 good
**Presentation:** 4 excellent
**Contribution:** 3 good
**Rating:** 8
**Confidence:** 4

**Summary:**

The paper critiques instruction tuning (IT) by showing that the IT models are not robust to rephrasing of the same prompt that were not seen in training. More surprisingly, using a prompt that 1) was seen in training, but 2) is incorrect for the context has a higher accuracy than an unseen but correct prompt. They propose a mitigation strategy by adding a soft prompt that is explicitly optimized to make the model see the different rephrases as similar to each other. They also release a dataset of 319 prompt rephrasings.

**Strengths:**

- This seems like a significant result overall, and explicitly shows an important weakness in IT models
- In general, this kind of work (exploring what "success on a benchmark" really means) is extremely important for the field. You are raising a really important issue with fine tuning, and this is impactful.
- The experiments in the paper are well-executed, well-presented, and broad across models and datasets.
- It's interesting to explicitly introduce the rephrasing invariance into the fine tuning data-- it's sort of an  equivalent of regularization, I think?
- Thanks for including figure 4! It helped a lot with the intuition in that section.

**Weaknesses:**

- nit: "instruction-tuned models are not especially robust to instruction rephrasings" → this is very clear, say it first. Better yet, make it the title :)
- I don't know if just releasing prompt rephrasing is a significant enough contribution-- https://openreview.net/forum?id=9Vrb9D0WI4 and others also have datasets of different rephrasing of prompts for the same task.

**Questions:**

- It looks like just fine tuning (or even fine tuning + KL) severely drops performance. Do you know why? This seems very surprising
- Why use NLP grad students to generate rephrasing of the prompts, when you could just use an LLM (which it seems like you do later for the eval set)?

---

> ### Author Response · Authors · 2023-11-20
> **Response to RNd8**
>
> We thank the reviewer for their thoughtful review, and we are glad that they found the analysis here interesting and potentially impactful.
>
> **W1 (Title of the paper)**: We thank the reviewer for the suggestion, and agree this is the main takeaway; we will emphasize this point earlier on (and consider using it as our title) in future version of the manuscript.
>
> **W2 (Rephrasing Contribution)**: We agree with the reviewer that the release of this data is, on its own, not a substantial contribution. However, we do think this dataset is novel with respect to existing resources (e.g., T0/PromptBench, Flan, and others) which have released rephrasings of instructions for the same tasks intended for model **training**. By contrast, the dataset of diverse instructions we release with this work is intended to aid model **evaluation**, specifically on tasks in two major benchmarks, MMLU and BBL. Our hope is that this resource will both permit reproducibility of our work, and facilitate additional research into the robustness of instruction-tuned models.
>
> **Q1 (Fine-tuning performance)**: Please see our general response, (1). Briefly, we think this is an instance of “catastrophic forgetting” and have provided new results that support this hypothesis in Appendix G.
>
> **Q2 (Rephrasing source)**: We were interested in evaluating the robustness of instruction tuned models to novel instructions, as they might be observed in deployment—this means we were interested not strictly in rephrasings but in novel instructions written anew by practitioners for specific tasks of interest. Collecting new instructions from NLP graduate students (who are familiar with instructing language models) better aligns with this envisioned stress-testing than using LLMs to automatically rephrase existing instructions; in particular this yields a sample from the distribution of instructions which might plausibly be written by practitioners, so we felt it best to manually collect these for evaluation.
>
> Note that in the proposed methods to make models more robust to instruction variations (Section 5), we did use LLMs to paraphrase existing instructions from the training data as a means to augment the data and realize the proposed objective term.

---

### Author Response · Authors · 2023-11-20
**General Response (To All Reviewers)**

We thank all reviewers for their thoughtful comments. Here we answer questions raised by multiple reviewers (in a separate comment we list edits to the revised manuscript).


### (1) RE: Performance degradation when fine-tuning (RNd8, KbYU)

A few reviewers highlight the somewhat curious result that simply (continuing) fine-tuning on newly collected instructions degrades performance. We agree that we should have investigated this further, and have now done so (Appendix G).

We hypothesize that is a result of “catastrophic forgetting”: The fine-tuning data is small relative to the original train sets, and to reflect the scenario of using instruction-tuned LLMs on new tasks/data, we collected paraphrases from tasks in the FLAN/Alpaca sets, but evaluated on MMLU/BBL. These have similar types (e.g., QA), but the distributions will naturally differ. Consequently, fine-tuning on these instances without additional objective terms may degrade benchmark performance. To evaluate this hypothesis we performed two new experiments.

**1. Loss During Fine-tuning (Fig. 9, Appendix G)**

**Procedure**: We continue to fine-tune Flan-T5-3B (instruction-tuned) and T5-V1.1-3B (not instruction-tuned) with the same data for 25 epochs and evaluate if this degrades instruction following capabilities. T5-V1.1-3B serves as a reference to verify if the instruction fine-tuning examples are generally helpful. We monitor loss on (a) a validation set sampled from the Flan collection (b) MMLU with unobserved/observed instructions.

For Flan-T5-3B (instruction-tuned) **loss** on both datasets **monotonically increases**; additional fine-tuning on this small set of instructions harms performance, supporting the “forgetting” hypothesis. By contrast, for T5-3B (not initially instruction-tuned), loss decreases, especially on (b), suggesting that the instructions are generally useful. Together, these results strongly suggest catastrophic forgetting due to the modest training set size, as hypothesized.

Next we investigate what happens to the representations in this setting; our hypothesis is that without the proposed alignment objective, fine-tuning with additional paraphrases does not bring equivalent instructions closer together.

**2. L2 Distances after Fine-tuning (Fig. 10, Appendix G)**

**Procedure**: We continue to fine-tune Flan-T5-3B with the same data for 25 epochs. On all BBL datasets and MMLU, we calculate average l2 distances between unobserved instructions and their nearest observed instruction over all examples (as in Table 6).

**Result**: The distance between unobserved instructions and their closest observed instructions nearly doubles to an average of 5.66, indicating that fine-tuning on augmented data without the proposed KL term (to align representations) actually makes equivalent instruction representations less similar, consistent with the observed degradation.

We have added these results to Appendix G.


### (2) RE: Variance/statistical significance of results (cGUUu, hcVs)

The high variances are due to our reporting aggregated results over many large benchmark tasks (see Appendix H for disaggregated results).

Concerning results in Table 3: A t-test pairing averages over the performances achieved using observed and unobserved instruction sets on the same model and dataset yields p << 0.001. We also run a paired t-test for each model (Flan T5 XL/XXL, Alpaca 7B/13B, and T0++) separately, finding p < 0.05 for all models except T0++ (p=0.13).

For improvement (Table 7), a t-test on paired results for the respective models corresponding to performance using (a) the standard fine-tuning objective and (b) the paraphrase alignment objective proposed in 5 also gives p < 0.05 for both FLAN T5 XL and Alpaca 7B (the only two models evaluated for this).

We have added these significance test results to Tables 3 and 7.


### (3) RE: Novelty of findings (cGUu, hcVs)

We acknowledge that this work builds on prior recent efforts, especially Webson and Pavlick (2022), as discussed in Section 2.

Our contribution is differentiated by our focus on instruction-tuned models and their robustness to appropriate (not adversarial) instructions, as might be encountered in use. Our main finding is that LLM performance degrades consistently when provided with appropriate instructions unobserved in training. This complements past work on more adversarial stress-testing, and highlights important practical weaknesses, calling into question what benchmarks really tell us (as noted by RNd8). Even if somewhat expected, our work provides robust empirical evidence and delves into underlying issues (the relationship between performance and similarities in representation space).

Finally, we have proposed and evaluated a new method to improve robustness which is directly informed by our analysis: explicitly align the representations of equivalent instructions. This yields modest but consistent improvements.

---

### Author Response · Authors · 2023-11-20
**Summary of Changes to Paper**

Informed by reviewer comments, we have revised our submission; we highlight the main changes in the manuscript here for convenience.

Main paper:
- **Table 3 on page 4**: added significance test results
- **Table 7 on page 9**: added significance test results

Appendix:
- We have slightly rearranged the order of the Appendices.
- **Appendix B.3 on page 16**: added more information (the length of the prefix, gradient accumulation, etc.) regarding implementation details.
- New Section: **Appendix B.3.2 on page 18**;  Ablation of the alignment effect of the proposed method.
- New Section: **Appendix F on page 22**; Added experiments exploring how the alignment process helps with in domain robustness.
- New Section: **Appendix F.1 on page 22**; Added experiments on robustness to negation.
- New Section: **Appendix G on page 23**; Results and discussion regarding Catastrophic Forgetting in Fine-tuning, explaining the observed performance degradation in this setting.
- New Section: **Appendix J.3 from page 37 to 38**; Disaggregate results of Table 7.

We have also updated the supplemental material to include the newest code for running the experiments we conducted during this response phase.

To ensure reproducibility of the newly reported results, we place all the scripts for the experiments under the directory “./rebuttal_scripts/”.

---

### Meta-Review · Area_Chair_3jps · 2023-12-11

**Metareview:**

The paper presents a study of the robustness of instruction tuned models to rephrasing of the instructions.
Reviewers generally liked the papers. They found the work to be impactful, revealing key weakness in Instruction tuned models and raising crucial questions about what success on benchmarks truly signifies. The experimental design was praised for its clarity, thoroughness, and relevance across a wide range of models and datasets. The study's approach, particularly its method for improving model robustness against varying instruction phrasings, was highlighted as another strength.
Reviewers also raised a few concerns including questions about novelty of the robustness experiments compared with prior work, variance and significance testing concerns, and lack of detailed explanations of the approach. Most of the concerns seem to be minor and addressed during the discussion period.

**Justification For Why Not Higher Score:**

While strong, in terms of potential impact and the presented ideas, the paper does not seem to be among the top 5% of all accepted submissions.

**Justification For Why Not Lower Score:**

All reviews were positive and I also liked the paper. This type of work might appeal well to a broad audience.

---

### Decision · Program_Chairs · 2024-01-16

Accept (spotlight)